**Investigation**

# Residues neighboring an SH3-binding motif participate in the interaction in vivo

David F. Jordan [ID] ,[1,2,3,4,*] Alexandre K. Dubé [ID] ,[1,2,3,4,5] Ugo Dionne [ID] ,[2,3,4,6,7,8]
David Bradley [ID] ,[1,2,3,4,5] Christian R. Landry [ID] [1,2,3,4,5,*]

[1]Département de biochimie, microbiologie et bio-informatique, Université Laval, 1045 Avenue de la Médecine, Québec, QC G1V 0A6, Canada
[2]Institut de Biologie Intégrative et des Systèmes (IBIS), Université Laval, 1030, Avenue de la Médecine, Québec, QC G1V 0A6, Canada
[3]PROTEO-Regroupement Québécois de Recherche sur la Fonction, l'Ingénierie et les Applications des Protéines, Université Laval, 1030, Avenue de la Médecine, Québec, QC G1V 0A6, Canada
[4]Centre de Recherche en Données Massives de l'Université Laval, Université Laval, 1065, Avenue de la Médecine, Québec, QC G1V 0A6, Canada
[5]Département de Biologie, Université Laval, 1045 Avenue de la Médecine, Québec, QC G1V 0A6, Canada
[6]Centre de Recherche du Centre Hospitalier Universitaire (CHU) de Québec, Université Laval, 11, côte du Palais, Québec, QC G1R 2J6, Canada
[7]Centre de Recherche sur le Cancer, Université Laval, 9, rue McMahon, Québec, QC G1R 3S3, Canada
[8]Present address:Lunenfeld-Tanenbaum Research Institute, Sinai Health, 600 University Avenue, Toronto, ON M5G 1X5, Canada

*Corresponding authors: David F. Jordan, Département de biochimie, microbiologie et bio-informatique, Université Laval, 1045 Avenue de la Médecine, Québec, QC G1V 0A6, Canada. Email: david.jordan.2@ulaval.ca; Christian R. Landry, Département de biochimie, microbiologie et bio-informatique, Université Laval, 1045 Avenue de la Médecine, Québec, QC G1V 0A6, Canada. Email: christian.landry@bio.ulaval.ca

In signaling networks, protein–protein interactions are often mediated by modular domains that bind short linear motifs. The motifs' sequences affect many factors, among them affinity and specificity, or the ability to bind strongly and to the appropriate partners. Using Deep Mutational Scanning to create a mutant library, and protein complementation assays to measure protein–protein interactions, we determined the in vivo binding strength of a library of mutants of a binding motif on the MAP kinase kinase Pbs2, which binds the SH3 domain of the osmosensor protein Sho1 in Saccharomyces cerevisiae. These measurements were made using the full-length endogenous proteins in their native cellular environment. We find that, along with residues within the canonical motif, many mutations in the residues neighboring the motif also modulate binding strength. Interestingly, all Pbs2 mutations that increase binding are situated outside of the Pbs2 region that interacts with the canonical SH3-binding pocket, suggesting that other surfaces on Sho1 contribute to binding. We use predicted structures and mutations to propose a model of binding that involves residues neighboring the canonical Pbs2 motif binding outside of the canonical SH3 binding pocket. We compared this predicted structure with known structures of SH3 domains binding peptides through residues outside of the motif, and put forth possible mechanisms through which Pbs2 can bind specifically to Sho1. We propose that for certain SH3 domain–motif pairs, affinity and specificity are determined by a broader range of sequences than what has previously been considered, potentially allowing easier differentiation between otherwise similar partners.

Keywords: short linear motif; SRC Homology 3 (SH3) domain; deep mutational scanning; binding affinity; structure prediction

## Introduction

Cells possess complex and robust signaling networks that detect stimuli and trigger responses. These signaling networks are composed of a series of protein–protein interactions, which are often mediated by modular interaction domains. Many classes of protein interaction domains are shared across pathways and species, yet fulfill different roles and functions (Pawson et al. 2002). The different constraints placed on interaction domains by their distinct roles contribute to explaining the divergence in sequence between homologs of the same domain (Ernst et al. 2010; Dionne et al. 2022). Therefore, to understand how domain sequences have evolved and continue to evolve, the phenotypic consequences of mutations must be understood. Two of the major phenotypes for protein–protein interaction domains are affinity, that is, the strength of binding to the partner protein, and specificity, that is, the ability to bind to the appropriate partner proteins,

and to not form spurious interactions with other proteins in the cellular environment (Ivarsson and Jemth 2019).

Many interaction domains bind short, intrinsically disordered stretches of their interaction partners, known as short linear motifs (SLiMs) or simply binding motifs (Gouw et al. 2018). Binding motifs are involved in a wide swath of cell signaling pathways, and underlie many important mechanisms in human and other cells (Kumar et al. 2024). Uncovering the determinants of binding can elucidate the functioning of human cells and certain diseases that affect them, such as various cancers in which motif-mediated signaling plays a part (Van Roey et al. 2014). Additionally, viruses also express their own binding motifs, disrupting the signaling machinery of their hosts (Davey et al. 2011). Understanding domain–motif binding can also allow motifs to be used as tools, for example, by attaching therapeutic proteins to domains and motifs to hydrogels, to ensure the gradual release of the therapeutic proteins in human tissues (Delplace et al. 2019).

Interactions mediated by domain–motif associations are relatively weak, with dissociation constants ($K_D$) in the micromolar range, while domain–domain binding typically results in dissociation constant values orders of magnitude smaller, indicating much stronger binding (Van Roey et al. 2014). While such low affinity could simply be caused by physical limitations, such as a smaller interface, adaptive hypotheses have been put forward which suggest that an overly strong affinity can have deleterious effects on the cell, for instance by compromising specificity (Haslam and Shields 2012; Karlsson et al. 2016). Alternatively, this low intrinsic affinity can be compensated by other factors. For example, certain protein–protein interactions depend on the simultaneous binding of many domain–motif pairs (Liao et al. 2020). In other cases, motifs bind outside of the canonical binding pockets of their partner domains, although this does not always lead to increased affinity (Douangamath et al. 2002). The regions surrounding a domain can also have an effect on motif binding preference, as inserting domains into new protein backgrounds can change the partners that the domain binds to (Dionne et al. 2021, 2022).

Binding affinity and specificity have previously been explored in the context of the interaction between binding domains and motifs, by measuring the binding of mutants *in vitro* (Yu et al. 1994; Zarrinpar et al. 2003; Tonikian et al. 2009; Vincentelli et al. 2015; Kazlauskas et al. 2016). However, the interaction of proteins in the cellular environment is a more complex situation, and many interactions detected in vivo are not detected in vitro, and vice versa (Kelil et al. 2017). Many factors can modulate binding in the cell, including colocalization of partners, expression in the same cell cycle phases, and contributions from sequences outside of the immediate binding domain and motif, as well as other proteins that can interact with one or both of the partners (Ivarsson and Jemth 2019; Dionne et al. 2022). Earlier studies have measured the effects of limited numbers of domain and motif mutants in vivo (Zarrinpar et al. 2003; Marles et al. 2004). More recently, techniques such as deep mutational scanning (DMS) have been used to study the impact of large libraries of mutations on protein stability and function *in vivo* (Fowler and Fields 2014). Protein complementation assays (Tarassov et al. 2008; Michnick et al. 2016) have also successfully been used to measure the binding strength of mutant libraries of artificially expressed constructs (Diss and Lehner 2018; Faure et al. 2022; Robles et al. 2023; Bendel et al. 2024). Other studies have even used genome editing in yeast to endogenously tag interaction partners, allowing the measurement of the *in vivo* binding strength of libraries of mutants (Dionne et al. 2021; Dibyachintan et al. 2025). A recent study showed the effect on binding of combining mutations in a PDZ (Kennedy 1995) domain and its binding motif (Zarin and Lehner 2024). These in vivo techniques could be used to determine how a large library of mutations can affect domain–motif interaction affinity.

One powerful model to study domain–motif interactions is the SH3 domain. These globular domains are composed of around 60 residues and bind different proline-rich motifs (Kaneko et al. 2008). The model yeast *Saccharomyces cerevisiae*'s proteome contains 27 different SH3 domains (Dionne et al. 2022), including one in the high osmolarity glycerol (HOG) signaling pathway protein Sho1 (Saito and Posas 2012). The SH3 domain of Sho1 is known to interact with a motif on the MAPKK Pbs2, and this interaction can be strengthened by exposing cells to osmotic stress (Maeda et al. 1995; Posas and Saito 1997; Saito and Posas 2012) (Fig. 1a). The Sho1–Pbs2 interaction is thus a powerful model to study domain–motif interaction strength, due to the potential for increasing or decreasing affinity through mutations and the simple method of inducing an increase in the interaction strength using osmotic stress.

To better understand the impact of mutations on affinity, we measured the binding strength of a nearly complete library of mutations in the Pbs2 binding motif, in vivo, using a Dihydrofolate reductase (DHFR) Protein-fragment Complementation Assay (PCA) (Tarassov et al. 2008; Michnick et al. 2016). We measured the binding strength between single-residue Pbs2 motif mutants and Sho1, the canonical partner of Pbs2. We find that all single mutations that increase the interaction strength are found outside the canonical binding motif, and that certain residues outside the binding motif interact with Sho1 outside of the canonical binding pocket, and even outside of the SH3 domain. We use this finding to propose a model of Sho1–Pbs2 binding where residues outside the canonical Pbs2 motif bind an additional binding pocket on Sho1. We also compare this proposed model to other known SH3–motif interactions involving binding outside the canonical motif.

## Materials and methods
### Growth conditions

*Escherichia coli* cells were grown in 2YT medium with shaking at 37 °C for liquid cultures. *S. cerevisiae* cells were grown in YPD medium, synthetic complete (SC) medium buffered to a pH of 6.0, or PCA medium (with or without 200 μg/ml methotrexate (MTX, Bioshop Canada) diluted in dimethyl sulfoxide (DMSO, Bioshop Canada), and 2.5% noble agar for solid plates, as specified. When needed, 200 μg/mL of G418 (Bioshop Canada), 100 μg/mL of nourseothricin (Nat, Jena Bioscience), or 250 μg/mL of hygromycin B (Hyg, Bioshop Canada) were used as antibiotic selection agents. Also, when specified, 1 M of sorbitol (Bioshop Canada) was added to either liquid or solid PCA medium. Liquid cultures were grown at 30 °C with agitation at 250 rpm. See Supplementary Table 4 in Supplementary File 2 for the complete composition of all growth media used.

### Cloning

All plasmids were constructed by Gibson assembly (New England Biolabs 2020). As a template for the *PBS2* DMS library construction, a pUC19 plasmid (Addgene #50005) was amplified and digested using SacI (New England Biolabs). This backbone was used for Gibson assembly along with an insert composed of a 248 base pair region of *PBS2* amplified from genomic DNA (pUC19-Pbs2). For CRISPR-Cas9-based genome editing, the pCAS plasmid (Addgene #60847) was modified as previously described (Ryan et al. 2016) to change the sgRNA to target either the Sho1-binding motif of Pbs2 (pCAS-Pbs2), the SH3 domain of Sho1 (pCAS-Sho1), the HPHNT1 cassette (pCas-HPH) (Dibyachintan et al. 2025), or the stuffer sequence, which was inserted into *PBS2* (pCAS-stuffer) (Dionne et al. 2021). The plasmids expressing fragments of *PBS2* were based on the pGD110 plasmid (Faure et al. 2022). For the two shorter *PBS2* fragments, primers were used to amplify part of the plasmid, with overhangs adding the *PBS2* fragment and homology to the plasmid backbone. For the two longer *PBS2* fragments, the fragments were amplified from genomic DNA, with primers adding homology to pGD110. In all cases, the pGD110 backbone was digested with HindIII (New England Biolabs), and Gibson assembly was used to add the *PBS2* fragments in 3′ of an open reading frame containing DHFR F[3] and a linker, followed by the *CYC* terminator. Proper plasmid assembly was confirmed using PCR and Sanger sequencing.

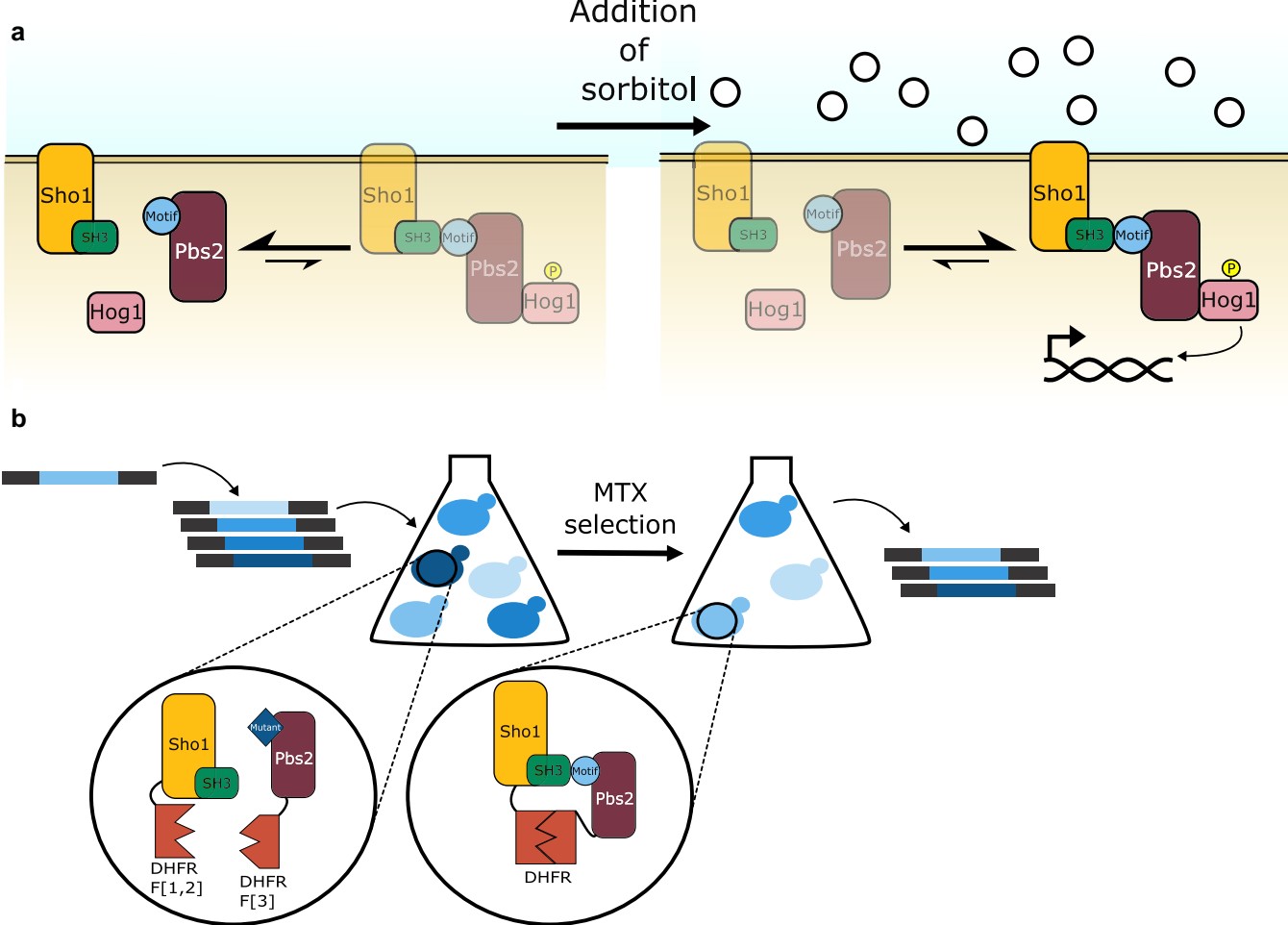

**Fig. 1.** The Sho1–Pbs2 interaction can be strengthened using sorbitol exposure, and the Dihydrofolate reductase (DHFR)-PCA can be used to measure the interaction strength between Sho1 and Pbs2. a) Simplified schematic of the HOG pathway showing Sho1, Pbs2, and Hog1 interactions under normal conditions (left) where they are largely separate, and during osmotic stress (e.g. with sorbitol, right) where pathway induction leads to complex formation. The resulting Hog1 phosphorylation leads to nuclear translocation and gene expression modulation. b) Schematic of the DHFR-PCA screen using a DMS *PBS2* mutant library. Single-residue variants of *PBS2* were inserted into the yeast genome at the native locus using CRISPR-Cas9 genome editing. Sho1 and Pbs2 were fused with complementary DHFR fragments. In cases where the Pbs2 variant does not interact with Sho1, the DHFR fragments remain separated. Upon interaction between a Pbs2 variant and Sho1, the DHFR fragments combine to reconstitute the functional DHFR and allow growth in the presence of methotrexate (MTX). More strongly interacting partners allow the cell to proliferate more rapidly (Freschi et al. 2013).

See Supplementary Table 5 in Supplementary File 2 for all primers used.

## Strain construction

For all strains used, see Supplementary Table 6 in Supplementary File 2. In all cases, proper strain construction was verified by PCR and Sanger sequencing. DHFR fragment strains were either taken from the Yeast Protein Interactome Collection (Horizon Discovery) (Tarassov et al. 2008), from a previously published work (Dionne et al. 2021), or constructed for this project using the same methods. Further deletions were done in DHFR fragment fused strains in order to knock out either *PBS2* or *SHO1*, using the *LEU2* cassette from pUG37 (Gueldener et al. 2002), through homologous recombination. This produced the following strains: *HOG1*-DHFR F[1,2]/*pbs2*::*LEU2*, *SHO1*-DHFR F[1,2]/*pbs2*::*LEU2*, *sho1*::*LEU2*/*PBS2*-DHFR F[3] and *sho1*::*LEU2*/*YBT1*-DHFR F[3]. This approach prevents the additional wild-type allele from influencing the measurements of interaction strength of the variants.

The Pbs2 mutants of interest were constructed from the Pbs2-DHFR F[3] fusion strain. First, codons 71 to 126 were replaced with a stuffer sequence (GGCGGAAGTTCTGGAGGTGGTGGT) that translates into a flexible linker sequence (GGSSGGGG), by co-transformation of the pCAS-Pbs2 plasmid, and of a repair template consisting of the stuffer sequence flanked by homology arms, following a previously published protocol (Ryan et al. 2016). This produced the Pbs2-stuffer-DHFR F[3] strain, which was used for the construction of the individually reconstructed mutants, for the construction of the DMS libraries, and also as a control to measure Pbs2 binding to SH3-containing yeast proteins. Individual mutants were reconstructed by co-transforming the Pbs2-stuffer-DHFR F[3] strain with the pCAS-stuffer plasmid targeting the aforementioned stuffer, and synthesized oligonucleotide sequences corresponding to the desired mutation (Integrated DNA Technologies) (Supplementary Table 7 in Supplementary File 2).

Three additional haploid strains were built for the extended motif DMS library (positions 85 to 100) and the individually reconstructed mutants used in the validation competition assays. The first 2 strains were Hog1-DHFR F[1,2]/Pbs2-stuffer-DHFR F[3] and Sho1-DHFR F[1,2]/Pbs2-stuffer-DHFR F[3]. These were built by adding the DHFR F[1,2] to either *HOG1* or *SHO1* in the MATα

Pbs2-stuffer-DHFR F[3] strain previously constructed through homologous recombination. The DHFR F[1,2] DNA fragments were amplified from plasmid pAG25-DHFR F[1,2]-linker-FLAG plasmid (Dionne et al. 2021). The third strain was *PBS2*-stuffer, which was used to measure the proliferation of mutants without the effect of the DHFR fragment. This was constructed in the haploid yeast strain BY4742, using the same strategy to insert the stuffer into *PBS2* as detailed above. The selected *PBS2* mutants for the validation assay were constructed in these 3 same strains by transformation with repair templates containing the mutations of interest, generated through fusion PCR. Briefly, forward and reverse primers were designed for each mutant to be constructed, which contained the mutated sequence instead of the wild-type sequence. These primers were used for separate PCR amplification of the *PBS2* sequence along with either common forward or reverse primers to create 2 overlapping fragments. These were combined in a second PCR, which fused the overlapping sequences together, to create a single fragment of 338 base pairs, which was identical to *PBS2* except for the mutated codon of interest. These fragments were transformed into the two DHFR F[1,2]/Pbs2-stuffer-DHFR F[3] strains and Pbs2-stuffer using the CRISPR-Cas9 strategy described above. Most mutations were constructed in all 3 strains, but a certain number could only be built in 1 or 2 of the strains. The successfully constructed mutants are listed in Supplementary Table 6 in Supplementary File 2. The Sho1 point mutants were constructed similarly to the Pbs2 mutants. First, a strain was constructed where *SHO1* was replaced by the *HPHNT1* cassette from pFA6-HPHNT1 (Janke et al. 2004) through homologous recombination. In parallel, mutant *SHO1* repair templates were constructed using fusion PCR with mutated primers, following the same strategy as the *PBS2* point mutations. This fusion PCR was amplified from genomic DNA where the *SHO1* locus was fused with DHFR F[1,2], and so the repair templates contained the DHFR F[1,2]. These repair templates were transformed into the *sho1*Δ::*HPHNT1* strain along with a pCas-HPH which targets the *HPHNT1* cassette (Ryan et al. 2016; Dibyachintan et al. 2025).

For the DHFR-PCA screen on solid media, the previously constructed wild-type Pbs2-DHFR F[3] and Pbs2-stuffer-DHFR F[3] were used. The SH3-containing proteins DHFR F[1,2] strains were obtained from the Yeast Protein Interactome Collection (Horizon Discovery). A version of Sho1 was also built where the entire SH3 domain was replaced by the same flexible stuffer used in the Pbs2-stuffer strain, using the same method as *PBS2*, but using the pCas-Sho1 plasmid.

The inducible SH3 expression strains for the growth curves with the Pbs2 fragments were constructed for this project following a previously published protocol (Lemieux et al. 2024). For this assay a previously constructed strain, AKD0678 was used, which contains the GEM expression system, where expression of the *GAL1* promoter is driven by a β-estradiol inducible artificial transcription factor (Aranda-Díaz et al. 2017), and where the coding sequence of *GAL1* is replaced by the stuffer sequence. This strain also contains the DHFR F[1,2] after the stuffer sequence. The SH3 domains of *SHO1*, *ABP1*, and *NBP2* were amplified and transformed into the stuffer, using the same strategy as for the *PBS2* mutants.

## Construction of DMS libraries

The initial DMS library covering the surrounding region of the *PBS2* motif Pbs2 (codons 71 to 126) was constructed as previously reported (Dionne et al. 2021). A region of 56 codons (168 nucleotides) was targeted, which corresponds to codons 71 to 126 of

the *YJL128C*/*PBS2* ORF. A degenerate oligonucleotide was designed for each codon to be mutated, with the 3 nucleotides of the mutated codon replaced by an NNN codon (see Supplementary Table 5 in Supplementary File 2 for all degenerate oligonucleotides used). This series of 56 oligonucleotides was used to amplify pUC19-Pbs2, in a two-step PCR procedure (Miyazaki 2011). The mutated plasmids were digested with DpnI (New England Biolabs, Inc. 2020) to remove residual template plasmid and then transformed into chemocompetent *E. coli* cells. Transformant colonies were resuspended in 5 mL of 2YT, and mutant plasmids were extracted using a miniprep plasmid extraction kit (Presto Mini Plasmid Kit, Geneaid) to obtain a plasmid library containing all the mutants. Prepared libraries were sequenced (see DNA sequencing section), to verify that the desired codon diversity was present. Once diversity was confirmed, the libraries were amplified from the plasmids and transformed into the *PBS2*-stuffer-DHFR F[3] yeast strain using the CRISPR-Cas9 strategy described in the strain construction section (Ryan et al. 2016). Mutants for all codons were mixed together in a masterpool, with 5 optical density units (OD) added for each codon. Frozen glycerol stocks were prepared for the individual suspensions of each codon, and multiple stocks were prepared for the masterpool. Genomic DNA was extracted from the masterpool using phenol/chloroform DNA extraction (Amberg et al. 2005), and as before, the library was sequenced to verify diversity. Transformed yeast libraries were stored at −80 °C. See Supplementary File 3 for complete details on library construction.

A second DMS library was constructed for the extended motif of *PBS2*, comprising codons 85 to 100. An oligonucleotide pool (Integrated Data Technologies, see Supplementary Table 8 in Supplementary File 2) was synthesized containing all possible NNK codons (K indicating either a G or T nucleotide) for the 16 codon region of interest. This pool was integrated into 2 different yeast strains: Sho1-DHFR F[1,2]/Pbs2-stuffer-DHFR F[3] and Hog1-DHFR F[1,2]/Pbs2-stuffer-DHFR F[3], by replacing the stuffer using the CRISPR-Cas9 strategy described in the strain construction section. As with the first library, the integrated libraries were sequenced to verify diversity. Transformed yeast libraries were stored at −80 °C.

## DHFR-PCA competition assays

The DHFR-PCA selection followed a previously published protocol (Dubé et al. 2022). Three liquid cultures were started, each from 100 µL of the masterpool containing all mutated codon positions, in SC complete medium. In the screen for the DMS library of the entire surrounding region (positions 71 to 126), these cultures were mated with either Sho1-DHFR F[1,2] or Hog1-DHFR F[1,2], and diploids were selected using nourseothricin and hygromycin B. A volume equivalent to 5 OD was spun down, and the supernatant was removed, to form a cellular pellet, which was stored at −80 °C. This is the initial time point used for sequencing. The selected diploid cultures were diluted to 0.1 OD/mL in 15 mL of liquid PCA medium with either 200 µg/mL of methotrexate with 1 M of sorbitol, the same concentration of methotrexate without sorbitol, or a control without methotrexate but with 1 M sorbitol. This PCA selection was grown at 30 °C for 96 h, except for the control cultures which were saturated after 24 h. After this first growth cycle, the cultures were diluted to 0.1 OD/mL into 15 mL of fresh liquid PCA medium, and a 5 OD pellet was spun down and stored at −80 °C. The second PCA cycle lasted the same amount of time, and at the end, as many 5 OD pellets as possible were prepared from each culture. Genomic DNA was extracted from the pellets of the second PCA selection culture using phenol/chloroform DNA

extraction (Amberg et al. 2005). The Pbs2 locus was amplified and sequenced, as described in the DNA sequencing section.

The liquid DHFR-PCA selection for the extended motif DMS library (positions 85 to 100) was done following the same protocol as the first DHFR-PCA selection, only ignoring the mating step, as the haploid strains already contained both DHFR fragments. However, a fourth control condition of PCA medium without methotrexate and without sorbitol was added. Three replicate pools were done for each growth condition. Frozen pellets were prepared as in the first DHFR-PCA selection.

The individually reconstructed validation strains were pooled by combining 100 μL of saturated overnight culture of each mutant in YPD. 6 pools were made for each condition, 3 pools each from 2 individual reconstructed colonies of each mutant. The same conditions were used as the second liquid PCA selection. For the cell proliferation measurements, SC medium with and without 1 M of sorbitol was used. Two cycles of selection were done, with 96-h cycles for the conditions with methotrexate and 24-h cycles for the conditions without methotrexate. Frozen pellets were prepared as for the previous liquid PCA selections.

## DNA sequencing and analysis

DNA sequencing was done in generally the same manner for all experiments, with minor changes in certain cases, as detailed below. DNA was extracted from libraries and frozen pellets using a standard phenol/chloroform genomic DNA extraction protocol (Amberg et al. 2005). The extracted DNA was amplified and barcoded using a Row-Column DNA barcoding strategy as previously described (Dubé et al. 2022). A summary of this strategy is included in Supplementary File 3. These barcoded pools were purified on magnetic beads and sent for sequencing on Illumina instruments. Unique combinations of barcodes were used to identify the DNA extracted from each pellet in each replicate of each condition. After preliminary analysis of the sequencing of the DNA from the pellets of the DHFR-PCA on the extended motif DMS library (positions 85 to 100), it was determined that more reads would be needed. Additional sequencing libraries were prepared fresh from the original DNA extractions. Instead of using a row-column barcoding approach, the PBS2 region was amplified using primers allowing the addition of Illumina Nextera barcodes (Illumina). These allow automated demultiplexing by the Illumina MiSeq instrument. All other steps were as described above.

The variant frequency in DMS libraries during construction and in the DHFR-PCA and proliferation screens was evaluated using custom scripts based on (Després et al. 2022). Python libraries pandas 1.5.2 (McKinney 2010), matplotlib 3.6.2 (Hunter 2007), and numpy 1.24.1 (Harris et al. 2020) were used for data manipulation and visualization. Quality was assessed using FastQC 0.12.1 (Andrews 2010). Reads were trimmed using Trimmomatic 0.39 (Bolger et al. 2014), then demultiplexed using bowtie 1.3.1 for the sequencing of the first DMS library (positions 71 to 126) (Langmead et al. 2009), Interstellar 1.0 following the RCP-PCR configuration for the first sequencing run of the extended motif DMS library (positions 85 to 100) (Kijima et al. 2023), and cutadapt 4.7 for the validation competition assay sequencing (Martin 2011). Forward and reverse reads were then merged using the PANDAseq software (Masella et al. 2012). Next, identical reads were grouped using vsearch (Rognes et al. 2016) and aligned to the wild-type PBS2 sequence using the needle function from the EMBOSS software (Rice et al. 2000). From this, the frequency of each variant was obtained.

From the frequency of each variant, the interaction scores and selection coefficients of each variant were calculated using custom R 4.3.1 (R Core Team 2023) scripts. The interaction score was calculated as the log-2-fold-change of the frequency of each variant at the end of the second cycle of methotrexate selection and the frequency of each variant after the selection for diploids (so immediately preceding the methotrexate selection). These log-2-fold-change scores were rescaled for each library, so that the median of nonsense codons was equivalent to −1 and the value of wild-type variants was 0. Variants were then filtered to keep only those that had 20 or more reads detected after diploid selection, in order to remove bias caused by low read counts leading to inflated scores. A unique score was calculated for each variant by taking the median score of all codons coding for the same amino acid variant in all replicates of the same condition. Scores were only kept for amino acid variants that had at least 3 replicates from any combination of codons not filtered out. For the competition assay of cell proliferation, a selection coefficient was calculated as in (McDonald 2019), with the number of generations calculated as the log-2-fold-change in optical density for each growth cycle.

We statistically verified which nonsense and silent mutants were abnormal using the R package rstatix (Kassambara 2021), by comparing every nonsense or silent mutation to all other nonsense or silent mutations (two-sided Mann–Whitney $U$ test, $P < 0.05$). The mutants which were significantly different were removed from the interaction score dataset. Similarly, we verified which mutants had defects in the control condition by comparing each missense mutant to the nonsense mutants (two-sided Mann–Whitney $U$ test with false discovery rate (FDR) corrected $P$-value > 0.05). The missense mutants, which were not significantly different from the nonsense mutants, were also removed from the interaction score dataset. We also removed from the interaction score dataset the mutants for which an abundance change was detected through gain or loss of interaction strength between Pbs2 and Hog1, as explained in the results section (two-sided Mann–Whitney $U$ test with FDR corrected $P$-value < 0.05). For details on the visualization of these results, see Supplementary File 3.

## DHFR-PCA growth assays

Growth was measured for individually reconstructed PBS2 mutants in PCA medium, to validate Pbs2 interaction strength measurements. DHFR F[3] fused strains with PBS2 mutations were mated with SHO1-DHFR F[1,2]/pbs2::LEU2, in order to produce diploids with complementary DHFR fusions. To mate these strains, 50 μL of precultures of each was mixed into 900 μL of YPD and incubated at 30 °C overnight without agitation; 2 μL of each mated culture was spotted on solid YPD + Nat + Hyg, to select for diploids. Exponential phase precultures of diploids were diluted to 0.1 OD/mL in 80 μL of PCA medium, in a 384-well plate, in 4 replicates, at 30 °C. Cultures were grown in four conditions: with 200 μg/mL of methotrexate and 1 M sorbitol, with only methotrexate, with only sorbitol, or with neither. The optical density of each well was measured every 15 min in a Tecan Spark plate reader (Tecan), for 72 h while incubating at 30 °C. The growth rates for SHO1 mutants were measured in the same way, except that the mutant strains were mated with either PBS2-DHFR F[3]/sho1Δ::LEU2 or YBT1-DHFR F[3]/sho1Δ::LEU2. The latter strain was used to measure Sho1 stability and abundance, as Ybt1 interacts constitutively with Sho1. In cases where the mutation destabilizes Sho1, the Ybt1 interaction will be affected, analogously to how Pbs2 destabilizing mutations affect the Pbs2–Hog1 interaction. The optical density was measured in 200 μL in a 96-well plate using an Agilent Biotek Epoch 2 plate reader (Agilent). To measure

the interaction of the *PBS2* fragments, the plasmids containing the *PBS2* fragments with the DHFR F[3] were transformed into 4 strains containing a DHFR F[1,2] at the *GAL1* locus: Sho1-SH3-DHFR F[1,2], Abp1-SH3-DHFR F[1,2] Nbp2-SH3-DHFR F[1,2] as well as a YFP-DHFR F[1,2] and a control wild-type strain. Three individual colonies were taken from each transformation as independent replicates and grown to exponential phase. The expression of the DHFR F[1,2] fused fragments was induced using 20 nM of estradiol, which drives expression of the *GAL1* promoter by the GEM artificial transcription factor, and should result in the same expression level of each construct (Aranda-Díaz et al. 2017). The growth rates were measured in the same manner as the *SHO1* mutants.

Growth curves data were analyzed using a custom script written in R (R Core Team 2023). The maximal growth rate was calculated using a custom function, by measuring growth over 5 time points, and taking the 98th percentile to avoid outliers. Pbs2 mutant P94R was disregarded, as growth of all replicates in the control condition in the absence of methotrexate was much lower than all other samples, suggesting a problem with strain construction. For the *PBS2* fragments, an interaction score was calculated by normalizing the growth rate using the growth rate of the strain measuring the interaction of the same SH3 domain with a DHFR F[3] fused to no Pbs2 fragment.

## Structure predictions and visualization

Structure predictions were done for wild-type Pbs2 and Pbs2 mutants in complex with Sho1 using AlphaFold 2.3.2 and the AlphaFold–Multimer implementation (Jumper et al. 2021; Evans et al. 2022). Default options were used, and only the top-scoring model for each mutant was relaxed. For Sho1, the sequence of the entire protein was used for prediction (Uniprot entry P40073). For Pbs2, residues 71 to 126 were used for the prediction (Uniprot entry P08018). Predicted structures were visualized using ChimeraX 1.8 (Pettersen et al. 2021). Contacts were predicted using the ChimeraX "contacts" tool. See Supplementary File 3 for details of contact analysis.

## DHFR-PCA on solid media against SH3 domain-containing proteins

The DHFR-PCA on solid media was based on previous work (Tarassov et al. 2008; Rochette et al. 2015). All colony manipulation was done using a robotic pin tool platform (BM5-SC1, S&P Robotics Inc.). The strains containing DHFR F[1,2] fused SH3-containing proteins were taken from the Yeast Protein Interactome Collection (Horizon Discovery) or built for this project. Colonies were cherry-picked from the 96-well plates and arrayed onto YPD + Nat plates for the DHFR F[1,2] strains or YPD + Hyg plates for the DHFR F[3] strains, in a 384 colony array. The DHFR F[1,2] strains were organized into a randomized pattern to avoid any effects caused by neighboring colonies. The 384 colony arrays were condensed into 1,536 colony arrays on YPD + Nat or YPD + Hyg, with the randomized DHFR F[1,2] pattern repeated 4 times. The outer 2 rows and columns of the 1,536 arrays were composed of LSM8-DHFR F[1,2] and CDC39-DHFR F[3], which grow well on PCA medium and serve to avoid any measurement bias from being on the edge of the array, for the interactions of interest. The 1,536 colony arrays were mated together by pinning the repeated DHFR F[1,2] array onto YPD plates and then pinning one of the DHFR F[3] arrays onto each plate. In this collection of arrays, each interaction between an SH3-containing protein and a Pbs2 variant was measured in 5 or 6 replicates. After 48 h of mating at 30 °C, the mated colonies were pinned onto YPD + Nat + Hyg plates to select for diploid cells. Two growth cycles of 48 h at 30 °C

on YPD + Nat + Hyg were done. After diploid selection, the plates were photographed using an EOS Rebel T5i camera (Canon), to verify that all colonies grew correctly. The mated arrays were then pinned onto solid PCA medium plates, either with or without methotrexate, and with 1 M of sorbitol. The arrays were grown for 2 cycles of 48 h at 30 °C in a custom growth and imaging platform (S&P Robotics Inc.), which incubated the plates and took a picture of each plate every 2 h.

These plate pictures were analyzed to measure the size of every colony. First, the ImageMagick command line tool (ImageMagick Studio LLC 2023) was used to crop the images of the selection plates as well as change them to grayscale and invert the colors. Colony sizes were then quantified using the Python package Pyphe (Kamrad et al. 2020). See Supplementary File 3 for complete details of command lines used. The colony sizes were analyzed and visualized using a custom script written in R 4.3.1 (R Core Team 2023), based on a previous analysis (Dionne et al. 2021). Colony sizes were filtered, and colonies which had not grown at the end of the diploid selection were removed from the PCA results. Colony sizes were log2 transformed and then normalized to avoid bias from a position in the array or from the plate they grew on. Scores were rescaled so that the median of the scores of the Sho1 and wild-type Pbs2 interaction was equivalent to 1 and the median of the scores of the Sho1 and Pbs2–stuffer interaction was equivalent to 0.

## Results
### Deep Mutational Scan of the region surrounding the Pbs2 binding motif reveals that few mutations modify binding to Sho1

The interaction between Sho1 and Pbs2 is modulated by an SH3 domain on Sho1 and the canonical binding motif on Pbs2, situated in positions 93 to 99, and which has the sequence KPLPPLP (Maeda et al. 1995). Previous computational work has suggested that the region surrounding binding motifs could play an important role in modulating binding, including in SH3 interactions and in particular in the Sho1–Pbs2 interaction (Stein and Aloy 2008; Kelil et al. 2016). To determine which Pbs2 residues play a role in binding, we undertook a Deep Mutational Scan (DMS) on 56 codons at positions 71 to 126, comprising the binding motif and its surrounding region. We generated a DNA library containing nearly all possible single codon mutations within this region and used CRISPR-Cas9 to insert it into the *PBS2* locus of S. cerevisiae, replacing the wild-type sequence (Dionne et al. 2021). The *PBS2* gene was fused with a Dihydrofolate reductase Protein-Fragment 3 (DHFR F[3]), and these cells were mated with cells containing a DHFR F[1,2] fusion with *SHO1*, creating diploid strains with complementary DHFR fragments on both interaction partners. Sho1–Pbs2 interaction reconstitutes a functional DHFR, which enables cell division in the presence of methotrexate (MTX) (Fig. 1b). Growth in the presence of MTX in a strain with both DHFR fragments thus serves as a proxy for the amount of Sho1–Pbs2 complexes forming, which depends on both the binding affinity and the local abundance of the interacting proteins (Tarassov et al. 2008 ; Freschi et al. 2013). Since the DHFR fusions are C-terminal, nonsense mutations in the library result in the absence of DHFR fragment expression.

To measure the interaction strength of the various mutants in the DMS library, the mutants were pooled, and DHFR-PCA competition assays were done in media containing both MTX and 1 M of sorbitol, in order to induce the HOG pathway (Ferrigno et al. 1998). The interaction score for each variant was calculated by

normalizing the log2-fold-change in variant frequencies before and after selection in MTX, determined by targeted sequencing of the PBS2 locus. Within each replicate, scores were normalized such that wild-type Pbs2 had a score of 0, and the median of nonsense mutants had a score of −1. Mutants that possess a higher interaction strength than wild-type Pbs2 therefore have a positive interaction score, while mutants that possess a lower interaction score than wild-type Pbs2 have a negative interaction score.

While many mutations within the canonical motif altered interaction strength, most mutations in the surrounding region (positions 71 to 126) did not affect Sho1 binding, except for a short section neighboring the canonical motif itself (Fig. 2a, Supplementary Fig. 1 and Supplementary Fig. 2 in Supplementary File 1, Supplementary Table 1 in Supplementary File 2). This finding supports the critical role of the motif and validates computational predictions that extended the binding interface to include nearby residues (Stein and Aloy 2008; Kelil et al. 2016). Since the effect on binding was strongly position dependent, we categorized different sections of Pbs2 as follows: the canonical binding motif is the conserved type I binding motif sequence situated in positions 93 to 99, while the extended motif is composed of the canonical motif and the neighboring residues which have a strong impact on binding, from positions 85 to 99.

DHFR-PCA signal reflects both binding affinity and protein abundance. To distinguish how mutations can impact these, we assessed the effect of Pbs2 mutants on the Hog1–Pbs2 interaction, which is independent of the SH3-binding motif (Murakami et al. 2008), thus serving as a control for Pbs2 local abundance. Therefore, mutations solely affecting Sho1 binding should not impact the Hog1-Pbs2 DHFR-PCA, whereas mutations affecting Pbs2 stability or local abundance will influence both the Sho1–Pbs2 and Hog1–Pbs2 assays.

We found that most mutations in the surrounding region of the Pbs2 motif had little effect on Hog1 binding (Fig. 2a). Furthermore, mutants that had a negative effect on Hog1-Pbs2 binding also had a negative effect on Sho1–Pbs2 binding, suggesting that the reduction in signal results from a loss of local abundance of Pbs2 (Supplementary Fig. 3 in Supplementary File 1). However, most mutations that affected Sho1 binding did not impact the Hog1 interaction, and therefore did not change the local abundance of Pbs2.

## Extended motif DMS library identifies mutations modulating interaction strength

Based on the initial results, we focused our subsequent analysis on the extended Pbs2 motif (codons 85 to 100), including position

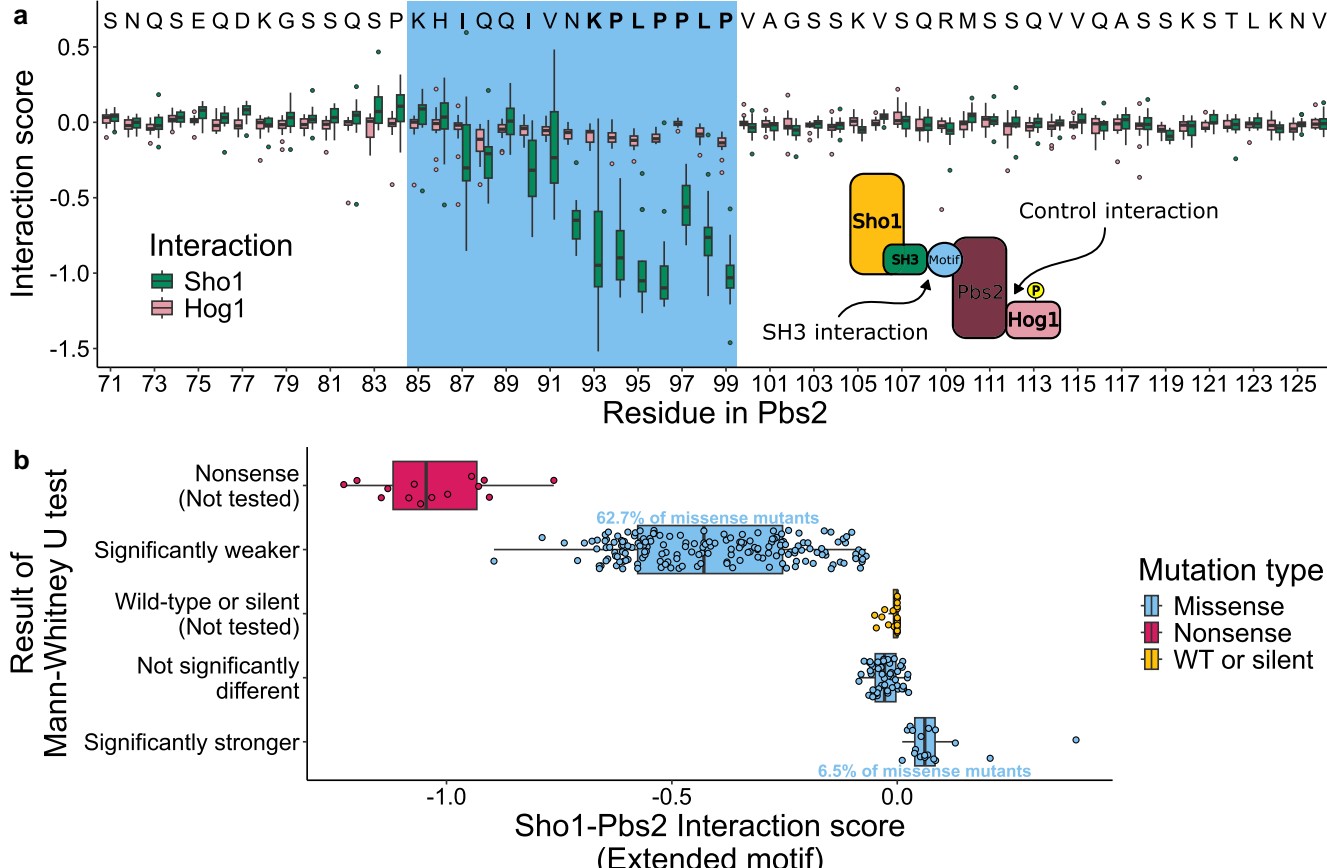

**Fig. 2.** Mutations in Pbs2 within and around the binding motif affect binding to Sho1. a) Interaction scores (normalized log2-fold-change after MTX selection) of Pbs2 mutations across the region surrounding the SH3 binding motif (positions 71 to 126), measured for interaction with Sho1 and Hog1. Wild-type Pbs2 residues are indicated above each position, with the extended motif (positions 85 to 99) highlighted in blue and the canonical motif (positions 93 to 99) residues in bold. Boxplots show the distribution of interaction scores for mutants at all positions (3 to 18 replicates per mutant; nonsense mutants excluded). b) Distribution of interaction scores for single amino acid mutants (3 to 18 replicates) within the Pbs2 extended motif DMS library (positions 85 to 100) when interacting with Sho1. Mutations are displayed on the y-axis, colored by type: nonsense (stop codon), WT or silent (synonymous), and missense (amino acid change). Missense mutants were categorized based on Mann–Whitney U test (FDR-corrected P < 0.05) against WT and silent mutations: Significantly weaker (negative score), Not significantly different, or Significantly stronger (positive score).

100 to measure potential effects immediately downstream of the canonical motif. The initial DHFR-PCA in a diploid strain led to incomplete measurement of Sho1–Pbs2 interaction, as some Pbs2 mutants presumably interacted with the Sho1 copy not fused with DHFR F[1,2]. Consequently, we built a new haploid strain containing both DHFR F[1,2] on Sho1 and DHFR F[3] on Pbs2. We also constructed a Hog1-DHFR F[1,2]/Pbs2-DHFR F[3] haploid strain to control for Pbs2 local abundance. These new strains capture all Pbs2 binding to Sho1 or Hog1, therefore allowing for more sensitive measurements. Using the same CRISPR-Cas9 approach as before, we introduced a second Pbs2 DMS library, comprising nearly all possible single mutations in Pbs2 positions 85 to 100, into the two haploid DHFR strains. Mutant interaction strengths were measured via DHFR-PCA as previously described (Supplementary Figs. 4 and 5 in Supplementary File 1). We then excluded 12 Pbs2 mutants showing significantly altered Hog1 interaction scores compared to wild-type and synonymous Pbs2 mutants (two-sided Mann–Whitney U test, FDR-corrected $P$-value < 0.05; Supplementary Fig. 6 in Supplementary File 1). This filtering step eliminated mutants affecting interaction through changes in Pbs2 abundance or stability, which influence both Sho1 and Hog1 interactions.

The impacts on the Sho1–Pbs2 interaction were varied (Fig. 2b, Supplementary Fig. 7 in Supplementary File 1, Supplementary Table 2 in Supplementary File 2). Only a small fraction of mutants exhibited interaction scores comparable to nonsense mutants, indicating that single mutations rarely abolish Sho1–Pbs2 binding completely. Comparison of missense mutant interaction scores to wild-type and synonymous variants revealed that many mutants displayed significantly stronger or weaker interactions (two-sided Mann–Whitney U test, FDR-corrected $P < 0.05$). Specifically, 18 mutants (6.5% of missense) showed significantly increased interaction, while 173 mutants (62.7% of missense) showed significantly decreased interaction compared to wild-type Pbs2. The impact of mutants that reduced the interaction scores was generally greater.

Given that osmotic stress enhances the Sho1–Pbs2 interaction (Maeda et al. 1995; Posas and Saito 1997 ; Saito and Posas 2012), we compared the interaction strengths of Pbs2 mutants with and without 1 M sorbitol (Supplementary Fig. 8 in Supplementary File 1). A strong correlation was observed between interaction scores under both conditions (Spearman's rho 0.98, $P$-value < 2.2×10$^{-16}$). As expected due to HOG pathway induction, the effects on binding were stronger in the presence of sorbitol. Under osmotic stress, strongly interacting Pbs2 mutants are even more often in contact with Sho1, increasing the number of reconstituted DHFR complexes, and further outcompeting weakly interacting mutants. Nevertheless, the high correlation suggests that Pbs2 mutants exhibit similar relative binding changes regardless of osmotic stress, and the fundamental impact of mutations is independent of HOG pathway activation.

To validate the pooled measurements, we individually reconstructed 24 mutants spanning the range of measured interaction scores in Pbs2-DHFR F[3] strains and mated them with the Sho1-DHFR F[1,2] strain. Growth rates of these diploids were then measured in DHFR-PCA medium. As with the competition assay, the growth rate is a proxy for interaction strength, which reflects affinity in the absence of a change in abundance. The growth curve results correlate strongly with the interaction scores (Spearman's rho 0.86, $P$-value 2.4×10$^{-6}$ with 1 M sorbitol, spearman's rho of 0.8, $P$-value 4.3×10$^{-6}$ without sorbitol), confirming the competition assay results (Supplementary Figure 9a in Supplementary File 1). To further validate the pooled data, we

performed a smaller-scale competition DHFR-PCA, using 67 individually reconstructed mutants with diverse interaction scores (Supplementary Fig. 10 in Supplementary File 1, Supplementary Table 3 in Supplementary File 2). This validation assay showed a strong correlation with the DMS library measurements (Spearman's rho 0.98, $P$-value < 2.2×10$^{-16}$ with 1 M sorbitol, Spearman's rho of 0.98, $P$-value < 2.2×10$^{-16}$ without sorbitol), thus validating the DMS library results (Supplementary Figure 9b in Supplementary File 1). To assess the direct impact of mutations on cell proliferation, we constructed strains with the same 67 mutations but lacking DHFR fragments. Cell growth and division thus only depended on the effect of the mutations, and not interaction strength. We then performed a pooled competitive growth assay in SC medium with 1 M sorbitol to measure the proliferation of each mutant. With the exception of nonsense mutations, no significant effect on cell proliferation was observed (Welch's *t*-test [FDR] corrected $P$-value < 0.05) (Supplementary Fig. 11 in Supplementary File 1, Supplementary Table 3 in Supplementary File 2). Notably, the strongest interacting mutant, I87W, shows a detectable, though not statistically significant, deleterious effect on cell proliferation.

## Structure prediction and Sho1 variants suggest a secondary contact between extended motif and Sho1

While the crystal structure of the Sho1-SH3 domain bound to a 9-residue Pbs2 segment shows the canonical motif (positions 93 to 99) occupying the entire canonical binding pocket (Kursula et al. 2008), our data reveal that many mutations outside this motif strongly affect binding as well. We hypothesized that residues in the extended Pbs2 motif (positions 85 to 99) might interact with Sho1 outside its canonical SH3 domain pocket. This is supported by NMR structures and studies of other SH3 domains showing that residues outside the canonical motif can modulate interaction strength and specificity (Feng et al. 1995; Rickles et al. 1995; Stollar et al. 2009; Takaku et al. 2010; Gorelik and Davidson 2012). Given the absence of experimental structures for full-length Sho1 or Pbs2, we used AlphaFold–Multimer predictions (Jumper et al. 2021; Evans et al. 2022) to explore potential interactions beyond the canonical binding site (Supplementary Fig. 12 in Supplementary File 1). Modeling the entire Sho1 protein along with the surrounding region of the Pbs2 motif (positions 71 to 126) revealed potential interactions outside of the canonical binding pocket of Sho1, mediated by residues neighboring the canonical binding motif of Pbs2 (Fig. 3a). The predicted structure shows a hydrophobic pocket formed by the Sho1-SH3 domain and a nearby loop, interacting with an alpha helix formed by Pbs2 residues 84 to 92 (Fig. 3b). Interestingly, all but one of the mutants in the extended motif which significantly increase interaction strength, are located within this predicted Pbs2 helix. Although the model also predicted proximity of some C-terminal Pbs2 residues to Sho1, mutations in these positions had minimal impact on binding (Fig. 2a), leading us to focus on residues within and near the extended motif.

To understand how specific mutations alter binding, we predicted the structures of the five Pbs2 variants exhibiting the strongest Sho1 interaction, using the same method as for wild-type Pbs2 (Supplementary Fig. 12 in Supplementary File 1). In particular, the strongest interacting mutant, I87W, is predicted to position its tryptophan side chain in close proximity (1.77 Å) to the non-canonical hydrophobic pocket, specifically within a subpocket formed by both SH3 and non-SH3 Sho1 residues. Furthermore, it is predicted to form a hydrogen bond with D333

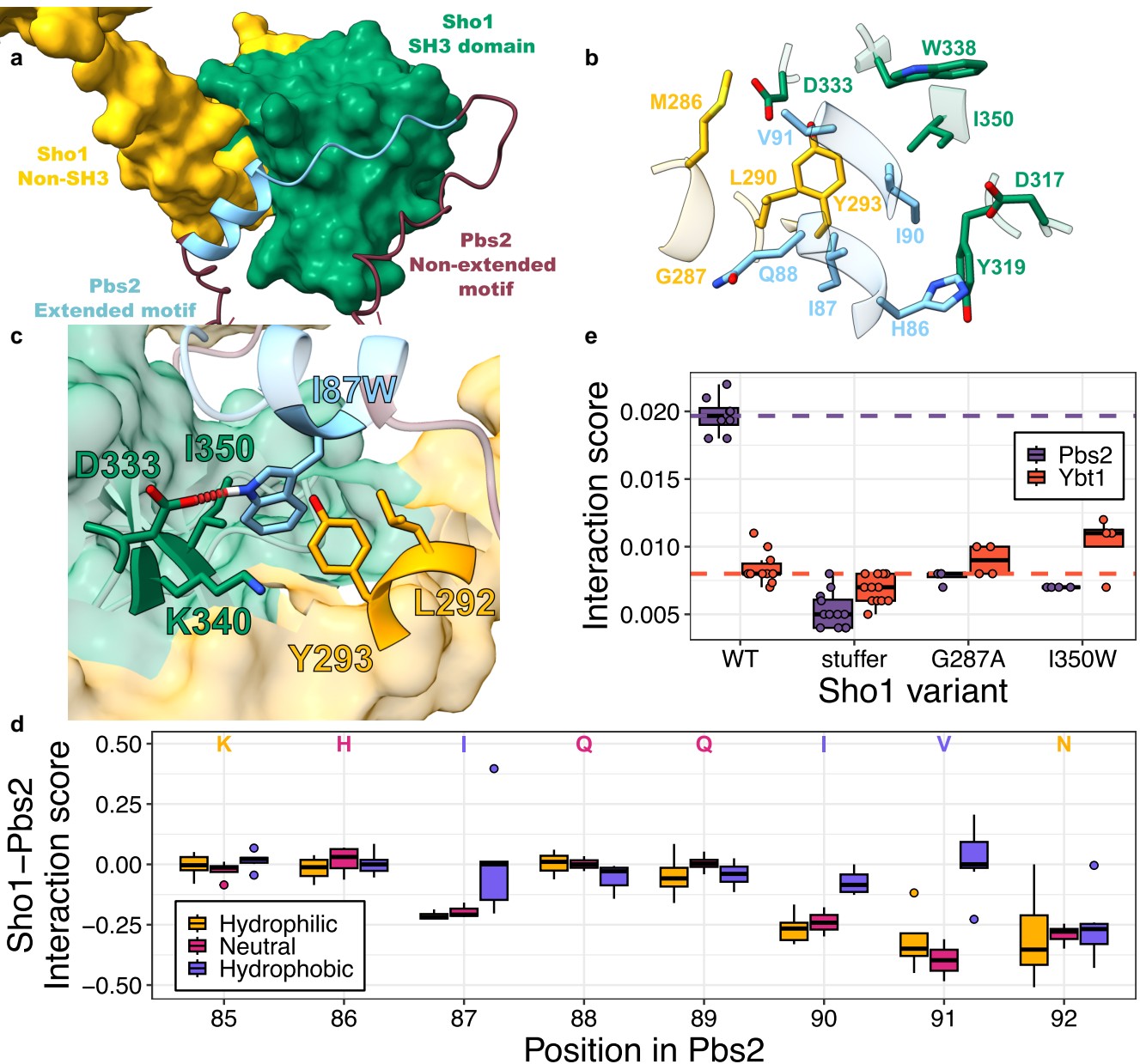

**Fig. 3.** Contacts mediated by positions outside the canonical binding motif of Pbs2 could modulate binding. a) Predicted structure of Sho1 in complex with Pbs2 residues 71 to 126, oriented to highlight the Pbs2 helix interacting with a hydrophobic pocket on Sho1, alongside the canonical motif (positions 93 to 99) in its binding pocket. For Sho1, the SH3 domain spanning positions 303 to 360 is colored in green, while the rest of the protein is in yellow. For Pbs2, the extended motif spanning positions 85 to 99 is in blue while the rest of Pbs2 is in dark red. b) Close-up view of the interaction between Sho1 and the extended motif (positions 85 to 99) of Pbs2. Only residues outside the canonical motif of Pbs2 which are in contact with Sho1 are shown, and the Sho1 residues with which they are in contact are also shown. The coloring of the residues is as in panel a, with Sho1 residues inside the SH3 domain in green and Sho1 residues outside the SH3 domain in yellow. c) Close-up view of the predicted Sho1–Pbs2(I87W) complex, showing the W87 side chain and a predicted hydrogen bond (red dotted line) with Sho1 D333. All other Sho1 residues with which the I87W residue is predicted to form contacts are shown with their sidechains. Coloring as in a). d) Boxplots showing interaction scores of Pbs2 mutants at selected positions (from DMS DHFR-PCA of the extended motif), grouped by the hydrophobicity of the substituted residue. Wild-type Pbs2 residue at each position is indicated and colored by its hydrophobicity (Hydrophobic: F, I, W, L, V, M, Y, C, A; Neutral: T, H, E, S, Q; Hydrophilic: R, K, N, G, P, D; Monera et al. 1995). Nonsense mutants are excluded. e) Interaction scores of select Sho1 variants with either Pbs2 or Ybt1, derived from growth rates from individual DHFR-PCA assays. Scores for individual replicates shown as points. The median wild-type Sho1 interaction score with Pbs2 (purple) or Ybt1 (orange) is shown with dashed lines. Stuffer indicates a Sho1 mutant where the SH3 domain was replaced by a neutral, flexible peptide (GGSSGGGG). The scores for all Sho1 variants are shown in Supplementary Fig. 14 in Supplementary File 1.

of Sho1 (Fig. 3c). These 2 factors potentially explain its strong interaction strength relative to all other Pbs2 variants. Similarly, the strongly interacting mutants H86W, V91L, and V91M are all predicted to place hydrophobic side chains within the non-canonical hydrophobic pocket, potentially contributing to

stronger interactions. For the remaining mutant, Q89D, the hydrophilic aspartate side chain is predicted to face away from the hydrophobic pocket, thus not disrupting these interactions. However, the mechanism of its increased binding strength remains unclear (Supplementary Fig. 13 in Supplementary File 1).

We hypothesized that replacing hydrophobic Pbs2 residues predicted to contact the hydrophobic non-canonical pocket with non-hydrophobic ones would disrupt this interaction and weaken binding. Consistent with this, analysis of mutations within the predicted alpha helix shows that replacing the initially hydrophobic residues at positions 87, 90, and 91 with non-hydrophobic residues reduces binding strength. Generally, substituting these positions with other hydrophobic residues had a less detrimental effect, and for V91, many hydrophobic substitutions even increased interaction strength (Fig. 3d). These results support the prediction that the side chains of residues at positions 87, 90, and 91 are located within the non-canonical hydrophobic binding pocket. This finding helps explain the strong interaction strength of Pbs2 mutant I87W, by showing the important role of hydrophobic interactions in position 87.

We further explored the presence of contacts between Pbs2 and surfaces outside the Sho1 canonical binding pocket. We introduced a series of Sho1 mutations in residues predicted to be in proximity to Pbs2, in strains containing the DHFR fragments, to measure whether these mutations would disrupt binding. We also built a Sho1 variant where the SH3 domain was replaced by a flexible, neutral "stuffer" sequence (GGSSGGGG), to measure the interaction strength without the contribution of the SH3 domain. We then assessed the effect of the mutations on interaction strength using growth assays. To control for Sho1 stability and abundance, similar to our approach with Hog1 for Pbs2 mutants, we measured the interaction of the Sho1 mutants with Ybt1, a protein previously found to interact with Sho1 independently of the SH3 domain (Dionne et al. 2021). We compared mutant growth rates to wild-type controls, looking for Sho1 variants that diminished Pbs2 binding but not Ybt1 binding (one-sided Student's *t*-test, FDR-corrected *P*-value < 0.05, Supplementary Fig. 14 in Supplementary File 1).

Contrary to our expectation, Ybt1 is not an ideal control, as the Sho1–Ybt1 interaction is weakened by the absence of the SH3 domain, contrary to a previous report (Dionne et al. 2021). Also, certain mutations seemingly increase the interaction strength between Sho1 and Ybt1, which could be due to stabilization of Sho1, an increase in affinity with Ybt1, or a change in binding equilibrium with Ybt1 caused by reduced binding to Pbs2. Still, in cases where the interaction of a Sho1 variant with Ybt1 is not significantly weaker than the interaction of wild-type Sho1 with Ybt1, we assumed that the local abundance of Sho1 was not diminished. Two mutations show a clear decrease in Pbs2 interaction, but no decrease in Ybt1 interaction: G287A and I350W (Fig. 3e). The first mutation is outside the SH3 domain, while the second is in the domain, but not in the canonical binding site. Both mutated residues are predicted to be within 5 Å of the Pbs2 extended motif. The G287 backbone is notably predicted to form a hydrogen bond with Pbs2 residue Q88, located in the predicted alpha helix. Together, our structural predictions and interaction measurements indicate that residues outside the canonical Pbs2 motif could directly interact with residues both within and without the Sho1-SH3 domain, thereby modulating binding affinity.

## Comparison of Sho1–Pbs2 binding to other SH3–motif pairs

Given the proposed role of the extended Pbs2 motif and the additional Sho1-binding pocket, composed of the SH3 domain and a nearby loop, in modulating interaction strength, we investigated if these features also contribute to the specificity of Pbs2 binding to Sho1. We reasoned that if this additional pocket were unique among yeast SH3 domains, it could enhance binding specificity, as the Pbs2 extended motif might be specifically adapted to this feature. First, we performed a multiple sequence alignment of yeast SH3 domains, including 25 flanking residues on each side, which encompasses all Sho1 residues predicted to be within 5 Å of Pbs2. Apart from key conserved residues within the SH3 domain itself, this analysis revealed no significant conservation of residues forming the additional Sho1 pocket (Supplementary Fig. 15a in Supplementary File 1). To assess potential structural conservation despite sequence divergence, we conducted a multiple structural alignment using MUSTANG (Konagurthu et al. 2006) on AlphaFold-predicted structures (Varadi et al. 2022) of yeast SH3 domains, with 25 neighboring residues. This analysis showed structural conservation only within the SH3 domains, with no significant structural conservation in the flanking regions, including those forming the additional Sho1-binding pocket (Supplementary Fig. 15b in Supplementary File 1). Therefore, we conclude that the additional binding pocket on Sho1 represents a unique structural feature among yeast SH3 domains.

Despite the lack of conserved structure of non-SH3 sequences, there are reports of 3 other yeast SH3 domains binding extended motifs: Abp1, the second SH3 domain of Bem1 (referred to as Bem1–2) and Nbp2 (Stollar et al. 2009; Takaku et al. 2010; Gorelik and Davidson 2012). We investigated whether these SH3 domains interact with the extended Pbs2 motif *in vivo* using a DHFR-PCA assay that measures colony growth on solid medium (Tarassov et al. 2008). The SH3-containing proteins were fused with DHFR F[1,2] and Pbs2 with DHFR F[3]. As a control, we used a Pbs2 variant where the extended motif (positions 85 to 99) and surrounding region (codons 71 to 126, the same positions as the initial DMS library) were replaced by the same stuffer peptide which was previously used to replace the Sho1-SH3 domain. The interaction strength of the SH3 domains was also measured against this Pbs2-stuffer variant.

Abp1 and Nbp2 interacted with Pbs2 at a similar strength to Sho1, while Bem1–2 showed no detectable interaction (Fig. 4a). However, Abp1 and Nbp2 also interacted with Pbs2-stuffer at comparable levels to wild-type Pbs2, indicating that their binding is independent of the Sho1-binding motif. Nbp2 is known to bind a second, distal motif on Pbs2 through its SH3 domain (Mapes and Ota 2004), while Abp1 likely binds Pbs2 through an alternative, unknown site. Screening all other known yeast SH3-containing proteins revealed significantly weaker interactions with both Pbs2 variants compared to wild-type Sho1 (one-sided Mann–Whitney *U* test, FDR-corrected *P*-value < 0.05; Supplementary Fig. 16 in Supplementary File 1). Furthermore, wild-type Pbs2 showed a stronger interaction with a Sho1 variant where the SH3 domain was replaced by the aforementioned stuffer (Sho1-stuffer), than Pbs2-stuffer with Sho1-stuffer, confirming the role of the non-SH3 portion of Sho1 in binding the extended motif of Pbs2 (Fig. 4a). Given that Sho1-stuffer abundance is similar to wild-type Sho1 (Dionne et al. 2021), these results demonstrate that the extended Pbs2 motif specifically binds to Sho1 through both its SH3 and non-SH3 regions.

To further test whether the Pbs2 motif and surrounding region can bind other SH3 domains, we used DHFR-PCA to measure the interaction of Pbs2 fragments of different lengths with the SH3 domains of Abp1, Nbp2, and Sho1. We expressed the SH3 domains at the same level to remove bias resulting from the different cellular abundances of Abp1, Nbp2, and Sho1. We also measured the interaction of the same Pbs2 fragments against a constitutively expressed YFP-DHFR F[1,2] to control for proper expression of the Pbs2 fragments (Lemieux et al. 2025). The canonical Pbs2 motif (positions 93 to 99) showed a significantly stronger interaction

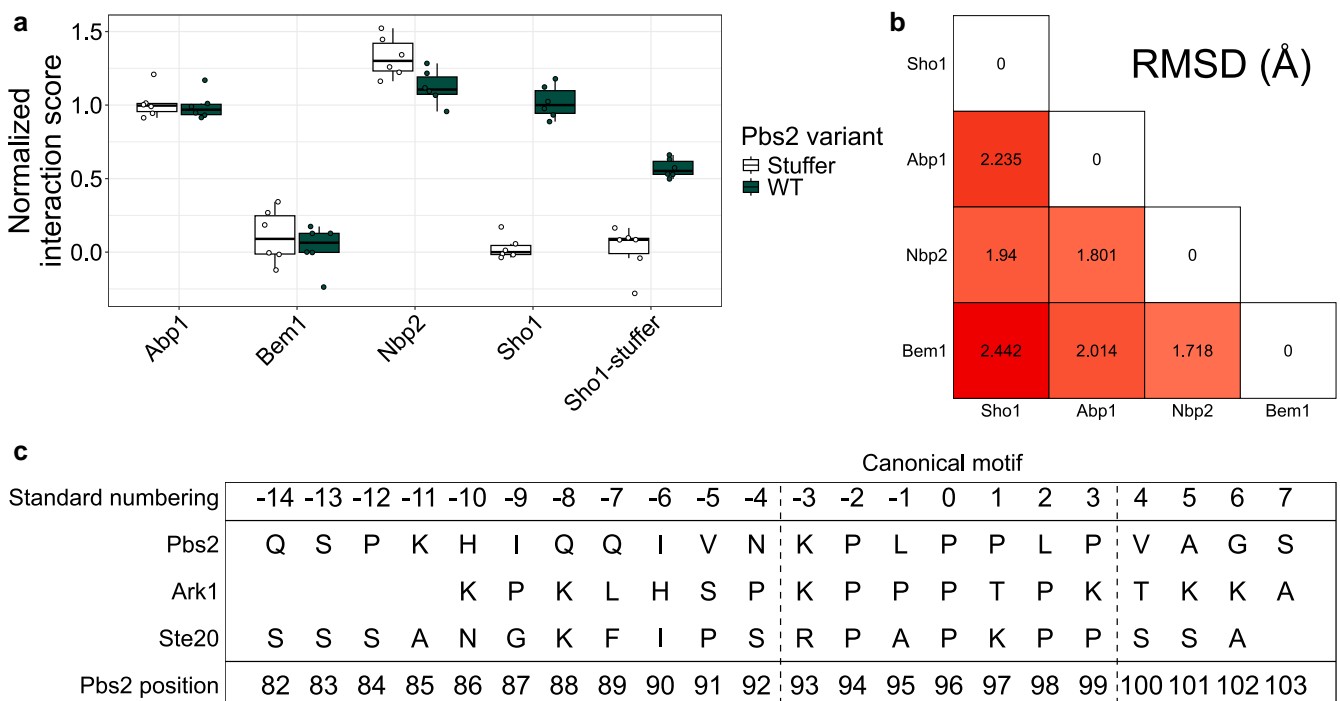

**Fig. 4.** The Pbs2 motif binds specifically to Sho1, despite similarities to other SH3 domains and motifs. a) Interaction score of wild-type Pbs2 (green) or Pbs2-stuffer (white) with select yeast SH3-containing proteins. All replicates shown as points. Scores are normalized relative to wild-type Sho1 interactions, with wild-type Pbs2 median at 1 and Pbs2-stuffer median at 0. The scores for all SH3-containing yeast proteins are found in Supplementary Fig. 16 in Supplementary File 1. b) Root mean square deviation (RMSD) of atom locations between matching backbone atoms in each pair of structures, as calculated using the R package bio3d (Grant et al. 2006). c) Sequences of Ste20, Ark1, and Pbs2, with standard numbering according to the motif, to compare equivalent positions (Lim et al. 1994). Ste20 and Ark1 positions included are those present in the structures used to generate panel b, with Pbs2 positions overlapping all represented positions.

(one-sided Student's *t*-test, FDR-corrected *P*-value < 0.05) with the Sho1-SH3 domain compared to the Nbp2-SH3 domain (Supplementary Fig. 17 in Supplementary File 1). This preference for the Sho1-SH3 domain was maintained and the interaction strength increased with 2 longer Pbs2 fragments: positions 85 to 100 (the region of the second DMS screen) and positions 69 to 124 (corresponding roughly to the Pbs2-stuffer replacement), which also showed significantly stronger binding to the Sho1-SH3 domain compared to the Abp1 and Nbp2 SH3 domains. While interaction scores with the non-Sho1-SH3 domains increased with Pbs2 fragment length, they were always weaker than the Sho1 interaction. This confirms the role of the extended motif and residues even more distant from the canonical site in maintaining binding specificity to the Sho1-SH3 domain.

To understand the basis of Pbs2 binding specificity to Sho1, we compared the predicted Sho1–Pbs2 structure with NMR structures of Abp1-Ark1, Bem1-2-Ste20, and Nbp2-Ste20, all involving peptides of similar length to the Pbs2 extended motif. The structures of the four SH3 domains and their bound peptides showed a conserved binding mode, with each peptide occupying the canonical SH3 domain pocket and extending toward the region corresponding to the additional Sho1-binding pocket (Fig. 4b, Supplementary Fig. 18 in Supplementary File 1). However, Abp1 and Nbp2 appear to interact with their respective extended motifs solely through the SH3 domain surface. In contrast, Bem1–2 utilizes a second, non-SH3 interface that forms a pocket with its SH3 domain, accommodating the extended motif, similar to Sho1. This Bem1–2 interface, identified as the Cdc42-interacting (CI) region (Takaku et al. 2010), exhibits no structural or sequence similarity to the corresponding region in Sho1. Furthermore, unlike Sho1, this

additional binding region in Bem1–2 is located C-terminal to its SH3 domain.

Beyond the conserved canonical motif positions, the extended motifs of Pbs2, Ark1, and Ste20 show limited sequence conservation. These extended motifs can be aligned using a previously developed standard numbering system where the first conserved proline is assigned position 0, preceding residues are numbered negatively, and following residues positively (Fig. 4c) (Lim et al. 1994). The binding specificity exhibited by Pbs2 may depend on the possible contacts between the extended motif and the different SH3 domains, as well as the non-SH3 surfaces in the cases of Sho1 and Bem1–2. By examining the interactions of these extended motifs with their respective partners, particularly focusing on residues within the extended motif but outside the canonical motif (positions 85 to 92), we aimed to identify determinants of Pbs2 binding specificity.

Three scenarios can differentiate Pbs2–Sho1 binding from the binding of other extended motifs to their partners: either contacts are present in Pbs2–Sho1 but absent in other pairs, contacts are absent in Pbs2–Sho1 but present in other pairs, or equivalent residues in extended motifs contact different residues of their interaction partner. We find examples of all 3 scenarios. For example, Pbs2 is predicted to contact Sho1 through a histidine residue in position −10, while Ark1, for example, does not have contacts so far from the canonical motif (Supplementary Fig. 19a in Supplementary File 1). Pbs2 is also missing certain interactions that are present in other pairs. For instance, in both Ste20 and Ark1, the residue at position −7 is crucial for binding, as in both cases, mutation of the residue to an alanine reduced binding affinity by at least an order magnitude, and this mutation had the largest impact of all

alanine substitutions outside the canonical motif (Stollar et al. 2009; Gorelik and Davidson 2012). In contrast, position −7 in Pbs2 is not predicted to interact with Sho1, though mutation to a hydrophilic residue increased interaction strength (Supplementary Fig. 19b in Supplementary File 1, Supplementary Table 2 in Supplementary File 2). Finally, the same position in different extended motifs can form distinct interactions with their partners. In position −6, the isoleucine in Pbs2 is predicted to contact Y319, W338 and I350 on the SH3 domain. In contrast, the histidine in position −6 of Ark1 forms a salt bridge with an aspartic acid on a loop of the Abp1-SH3 domain (Supplementary Fig. 19c in Supplementary File 1). The structural alignment of SH3 domains indicates that Sho1 has a gap at the position of the aspartic acid that forms the salt bridge (Supplementary Fig. 15b in Supplementary File 1, position 112). Even beyond these limited examples, the four extended motifs show differences in how they bind their partners. This suggests that the non-conserved regions within extended motifs enable a much greater degree of binding differentiation than the conserved canonical motif alone.

## Discussion

Binding motifs face the double challenge of binding their partner with appropriate strength while avoiding spurious interactions with non-partner proteins. To understand this balance in the Sho1–Pbs2 interaction, we used deep mutational scanning of the Pbs2 motif and its flanking regions to quantify the impact of single-residue mutations on Sho1 binding. Using structure predictions and mutations in both Pbs2 and Sho1, we conclude that binding occurs between Sho1 and Pbs2 outside of the canonical motif and the SH3 domain. We then compare the predicted Sho1–Pbs2 complex to existing structures of other SH3 domain-peptide complexes. Based on these comparisons, we propose several mechanisms by which the Pbs2 extended motif achieves specific binding to Sho1 over other SH3 domains.

One important consideration in domain–peptide interactions is the potential tradeoff between interaction affinity and specificity. Intuitively, a domain that can strongly bind its target motif might also exhibit increased unwanted interactions with similar motifs within the same family, due to their physical and chemical resemblance. This idea is supported by previous studies on SH2 and PDZ domains, which, like SH3 domains, bind short motifs. These studies have suggested that mutants exhibiting higher affinity for their target tend to display lower specificity, implying a potential incompatibility between high affinity and high specificity (Ernst et al. 2010; Haslam and Shields 2012; Kaneko et al. 2012; Karlsson et al. 2016). However, a computational study of binding motifs, including SH3 binding motifs, proposed that increased affinity can correlate with increased specificity (Kelil et al. 2016). Other works have emphasized the importance of the residues surrounding the canonical binding motif in determining both affinity and specificity (Li 2005; Ivarsson and Jemth 2019). In fact, one study computed that in SH3 interactions, nearly 30% of binding energy from the binding motif was contributed by residues outside the canonical motif, and suggested that the residues outside the canonical motif mostly play a role in determining specificity (Stein and Aloy 2008). Notably, our deep mutational scanning revealed that all mutations which significantly increase the interaction strength between Sho1 and Pbs2 are situated outside of the canonical motif. This suggests that the wild-type canonical motif might already be optimized for maximal binding strength. Therefore, we propose a model where the canonical motif residues in positions 93 to 99 are optimized and thus structurally

constrained for affinity, while the neighboring residues are less constrained and able to modulate affinity and specificity when mutated.

We show that the modified binding strengths from mutations outside the Pbs2 canonical motif arise from interactions with surfaces beyond the SH3 domain's canonical pocket, even involving non-SH3 surfaces on Sho1. Non-binding pocket interactions have previously been observed in multiple SH3-mediated interactions (Lee et al. 1996; Dalgarno et al. 1997; Li 2005; Stollar et al. 2009; Takaku et al. 2010; Gorelik and Davidson 2012; Gaussmann et al. 2024), and we find that at least one mutation outside the Sho1-SH3 domain can disrupt binding, suggesting that the mutated residue plays a role in the interaction. Our AlphaFold-Multimer predictions support this, revealing a hydrophobic pocket formed by the Sho1-SH3 domain and a non-SH3 loop, which residues of the Pbs2 extended motif can occupy. While the disordered nature of the Pbs2 flanking region suggests a dynamic interaction, these models provide a plausible mechanism for non-canonical motif-mediated binding. By removing the need for affinity to be dependent only on canonical binding motifs, the residues that surround motifs could bind to protein surfaces which are less conserved among SH3 domains, and therefore easier to differentiate. The presence of less conserved components, such as extended motifs or non-canonical binding pockets, could thus contribute to specificity among domains of the same family. The Sho1 hydrophobic pocket, formed by both SH3 and non-SH3 elements, exemplifies a unique structural feature that might differentiate Sho1, enabling Pbs2 to achieve higher affinity without necessarily increasing binding to other SH3 domains. This binding model, involving a larger number of residues in tuning affinity and specificity, could potentially uncouple affinity and specificity, avoiding a tradeoff between both traits. While affinity and specificity might be intrinsically coupled within the highly conserved canonical binding interface, the less constrained surrounding residues offer a greater capacity for independent variation of these traits.

SH3 domain-mediated interactions are known to be context-dependent, influenced not solely by the domain itself but also by the surrounding protein environment. This context includes the non-SH3 regions of the protein, interactions with other proteins, and cellular localization (Dionne et al. 2022). Consequently, an SH3 domain's interaction profile is not solely determined by its amino acid sequence. Indeed, studies show that relocating an SH3 domain to a different protein or expressing it in isolation significantly alters its interaction profile (Dionne et al. 2021; Lemieux et al. 2024; Dibyachintan et al. 2025). The observed effects of Pbs2 mutations on binding strength may be mediated not only through direct SH3 domain interactions but also within the broader Sho1 protein context. The interaction of the Pbs2 extended motif with non-SH3 Sho1 residues likely contributes to this context-dependent modulation. Other causes could include stereochemical adjustments of SH3 binding loops or changes affecting Sho1's interactions with other binding partners. Other proteins involved in the osmotic stress response, namely Ste11 and Ste50, have been shown to interact with the Sho1-SH3 domain outside the canonical binding pocket (Zarrinpar et al. 2004; Tatebayashi et al. 2006). Mutations in the Pbs2 peptide could potentially influence the stability of these interactions as well.

Previous studies have investigated the binding of Pbs2 motif variants (Zarrinpar et al. 2003), and our findings corroborate some of their observations. For instance, we also find that mutations near the canonical Pbs2 motif can either increase or decrease binding strength. This study also reported reduced fitness

for a promiscuous Pbs2 double mutant (Zarrinpar et al. 2003). In contrast, our analysis revealed no substantial fitness loss for single mutants, except for the I87W variant, which exhibited a noticeable, though not statistically significant, decrease in fitness (Supplementary Fig. 11 in Supplementary File 1). Interestingly, I87W is also the mutant showing the greatest increase in interaction strength, potentially explaining its slight fitness cost. It has previously been suggested that affinity increasing mutants may have deleterious effects on cell homeostasis, by reducing the dynamicity of the pathway, and impeding dissociation of the partners (Wang et al. 2018). Consistent with this, prior work on the Sho1–Pbs2 interaction demonstrated that a Sho1-SH3 domain mutation that increased affinity also led to a minor reduction in cell growth (Marles et al. 2004), indicating that the wild-type affinity may already be optimized, and that stronger binding may disrupt signaling.

In conclusion, we show that mutations in positions neighboring the canonical binding motif of Pbs2 can both increase and decrease binding strength with Sho1. We propose that these non-canonical Pbs2 residues can modulate binding affinity through interactions outside the conserved SH3 binding pocket. We hypothesize that Sho1 is similar to certain other yeast SH3 domains, which bind longer motifs through additional binding pockets (Stollar et al. 2009; Takaku et al. 2010; Gorelik and Davidson 2012) and propose mechanisms through which an extended binding motif could better differentiate binding to different SH3 domains. Moving forward, studies of SH3–motif interactions should encompass a larger range of residues, considering both the SH3 domain surface beyond the canonical pocket and the regions surrounding the motif. Our findings, along with previous research, highlight that in vitro studies using isolated domains and minimal peptides may not fully represent the complexity of domain–motif interactions within their native protein context. By understanding and harnessing the potential of extended motif binding, researchers may be better able to understand the impact of mutations in motifs on cell signaling and disease, and better able to use motif binding in technical and medical applications.

## Data availability

Strains and plasmids are available upon request. All analysis scripts and visualization scripts are available on Github at the following address: https://github.com/Landrylab/Jordan_et_al_2024. All data, including demultiplexed sequencing data, results from the DHFR-PCA experiments, growth curve data, predicted structures, protein alignments, and all files and information necessary to run the scripts are available in the following Dryad repository: https://doi.org/10.5061/dryad.79cnp5j3z. The Supplementary Files are available in the following Zenodo repository: https://doi.org/10.5281/zenodo.15721636. Results from the DHFR-PCA experiments on mutant libraries are also available as Supplementary Tables 1 to 3 in Supplementary File 2. Supplementary File 1 contains Supplementary Figs. 1 to 19, Supplementary File 2 contains Supplementary Tables 1 to 10, and Supplementary File 3 contains the Supplementary Methods.

## Acknowledgments

The authors thank Philippe C. Després, Moïra Dion, and Dan Evans-Yamamoto for help with sequencing data analysis. We also thank all members of the LandryLab team for thoughtful comments on the experiments and on figure conception. All molecular graphics were performed using UCSF ChimeraX 1.8 (Pettersen et al. 2021). The protein structure predictions were

enabled by computing resources provided by the Digital Research Alliance of Canada. The Saccharomyces Genome Database was used throughout the project (Engel et al. 2025).

## Funding

This work was supported by grants from the Canadian Institutes of Health Research numbers 387697 and 525827, and Human Frontier Science Program research grant RGP34/2018 to C.R.L. C.R.L. holds the Canada Research Chair in Cellular Systems and Synthetic Biology. D.F.J. was supported by the Canada Graduate Scholarship—Master's program from NSERC, a graduate scholarship from PROTEO, a graduate scholarship from the NSERC CREATE program EvoFunPath, a Postgraduate Scholarship—Doctoral from NSERC, and a Doctoral scholarship from FRQS. D.B. was funded by the EMBO Long-Term Fellowship (LTF) (ALTF 1069-2019). U.D. was funded by an NSERC graduate fellowship.

## Author contributions

D.F.J., A.K.D., U.G., D.B., and C.R.L. conceived and designed the study. C.R.L. supervised the work and obtained funding. D.F.J. and A.K.D. built the strains and performed the experiments. D.F.J. performed the computational analyses. D.F.J. wrote the first draft of the manuscript. D.F.J., A.K.D., U.G., D.B., and C.R.L. contributed to revising the final version of the manuscript.

## Conflicts of interest

None declared.

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

*Editor: A. Gasch*