## [Peer Review File · Genetics]

Residues Neighboring an SH3-Binding Motif Participate in the Interaction In Vivo

David Jordan, Alexandre Dubé, Ugo Dionne, David Bradley, and Christian Landry

NOTE: The reviews and decision letters are unedited and appear as submitted by the reviewers.

In extremely rare instances and as determined by a Senior Editor or the EIC, portions of a review may be redacted. If a review is signed, the reviewer has agreed to no longer remain anonymous.

The review history appears in chronological order.

Review Timeline:

Submission Date:	2024-05-15
Editorial Decision:	2024-07-08
Resubmission Received:	2024-11-07
Editorial Decision:	2024-11-27
Resubmission Received:	2025-06-13
Editorial Decision:	2025-06-30
Resubmission Received:	2025-07-23
Accepted:	2025-07-26

July 8, 2024

GENETICS-2024-307083

Residues Neighboring an SH3-Binding Motif Participate in Determining Affinity and Specificity In Vivo

Dear Christian,

First, I apologize for the long review time for your manuscript. There was a significant delay finding an appropriate internal (or external) AE and in the end I handled the role myself.

Three reviewers assessed your manuscript, and I have read it as well. While your manuscript is not currently acceptable for publication in GENETICS, we would welcome a substantially revised manuscript. You can read their reviews at the end of this email.

Both Reviewer 1 and 3 had significant questions and concerns about claims related to specificity changes of Pbs2 mutations. Reviewer 1 questioned interpretation of the scores, and Reviewer 3 raised a series of points that are important to address, both in terms of the assay in general and independent validation beyond a single impactful mutant. It will be important to address these points, along with a response to all of the reviewers points. The manuscript will likely be sent back out for re-review.

We look forward to receiving your revised manuscript. Please let the editorial office know approximately how long you expect to need for revisions.

Upon resubmission, please include:

1. A clean version of your manuscript;
2. A marked version of your manuscript in which you highlight significant revisions carried out in response to the major points raised by the editor/reviewers (track changes is acceptable if preferred);
3. A detailed response to the editor's/reviewers' feedback and to the concerns listed above. Please reference line numbers in this response to aid the editor and reviewers.

Additionally, please ensure that your resubmission is formatted for GENETICS

<https://academic.oup.com/genetics/pages/general-instructions>

Follow this link to submit the revised manuscript: Link Not Available

Sincerely,

Audrey Gasch
Senior Editor
GENETICS

Approved by:
Howard Lipshitz
Editor in Chief
GENETICS

Reviewer #1 (Comments for the Authors (Required)):

Jordan et al. perform deep-mutational scanning of a SH3-binding motif, along with its surrounding regions, and ask whether the mutations affect affinity to its cognate SH3-domain and to other SH3 domains. They use a well-validated fragment-complementation assay to form their conclusions. They also look briefly at protein-structural predictions to make sense of their results. Overall, I believe the experimental work to be of high quality and the results to be of sufficient interest to readers. However, I am not sure I fully agree with the interpretations of all their experiments (see point #2). I would also have liked to see some evolutionary analysis of the implications of their results, given the fact that there are many SH3-binding motifs known to interact with SH3 domains. There is also room to analyze more thoroughly the structural bindings of their SH3 peptide to other SH3 domains. Here are some suggested points to address:

1) The authors perform the experiment twice with slight modifications (comparing the full vs the extended motif). The reasons for this are well-motivated and in general the results between both experiments agree. There are exceptions, however, specifically in the regions around the KPLPPLP motif (position 86 to 89). Those change from -ve to +ve within a screen. What is the explanation for this? Is this due to some normalization to the mean instead of to the mean of WT sequences? I don't see how an

extra copy of SHO1 can make these regions deleterious in the initial screen and I don't understand why the WT sequences don't all have an interaction score of 0 in the repeat experiment. I see that the interaction 'score' is assay dependent, as in it depends on what else is being measured. I find this score makes it difficult to compare all the different assays. Perhaps consider normalizing to the WT sequence instead of the average sequence.

2) I'm not sure if I fully agree with the analysis of changes in specificity. The PCA assay measures affinity, and the Pbs2_Q82W mutant increases its affinity to SHO1: its "interaction score" is >1. That same mutant has an "interaction score" with Fus1, Cdc25 and SDC25_17W of less than 0.5. So, it seems the mutant's affinity to those SH3 domains have increased (which the authors claim is a loss of specificity), but the affinity to those domains has increased correspondingly less than the increased in affinity to SHO1. For me, this is still potentially an increase in specificity and it might change how one discusses the story of adaptation with respect to specificity/affinity. That is, specificity should be something about the balance in the changes in affinity between the 'true' target.

There are some other potentially interesting questions with the domains that have 'off-target' interactions lost/gained with some mutations. Some domains appear multiple times, is there any reasons? Are these domains more similar/dissimilar to SHO1's SH3 domain?

3) Only 2/10 mutations tested are inside the core KPLPPLP motif (position 93 to 99). I don't know if there is statistical evidence for the statement that there is a position dependent effect.

4) The interaction scores should be provided as a supplementary table.

5) I am not a seasoned structural biologist, but the structure for the Pbs2 extended sequence motif is shown in a strange orientation. I can't fully grasp how even a loop region could fold this way. Currently, it looks like 5-10 amino acids are tangled in a knot, which I would argue is not likely to be a sensible AlphaFold prediction. This poses some questions because the extended motif does not really have any amino acids that increase/decrease interactions but it does look like it forms extended contacts with the structure (as portrayed).

It also seems like there are some low hanging-fruit analysis here with some of the other SH3 domains (particularly the ones that appear multiple times in figure 3C). Do the mutations that increase/decrease specificity ALSO have a structural basis?

6) Some of the stories on evolution of motifs are compelling, but I found myself needing to look at alignments to see if this is born out. Indeed, the Pbs2 motif has strong evolutionary conservation in the extended motif. I89, for example, is barely impacted by the mutations on the PCA but is much more conserved than I87 where mutations to hydrophobic amino acids are observed. Why is that? Are other SH3-binding motifs also using the N-terminus of their motif to 'extend' specificity? There is some suggestions that this might be the case, at least with the SH3 'domain-ome' experiments where some mutations in the extended motifs alter binding to other SH3 domains.

7) Figure 4 B/C should be revised to have some form of concordance on the directions for 'interactions gained/lost'.

Minor comments:

1) Is the YPD really made with 2% Tryptone and not Peptone?

2) Line 296: small typo (WHen instead of When).

Reviewer #2 (Comments for the Authors (Required)):

In this manuscript, the authors investigate the long-suggested trade-off between affinity and specificity in protein-protein interactions in complex signaling pathways. They show through a series of mutational experiments, that regions outside the core motif in Pbs2 contribute to affinity to SH3 domain of Sho1, and labelled them extended motif. They showed that the majority of residues in the core binding motif are already optimized for binding to SH3, whereas residues in the rest of the extended motif can be mutated to improve affinity. They followed this with mutational scanning to investigate whether the improved or reduced affinity mutations introduced in the extended motif, have an opposing effect on specificity, and found no correlation between gain of affinity and loss of specificity. They finally use computational protein modelling to investigate some of the improved affinity mutations, and found that increased hydrophobicity in certain residues in the core domain motif allow better interaction with the hydrophobic pocket formed in the SH3 and adjacent region.

Overall, the manuscript is well-written, and experiments are well performed and interpreted. Few suggestions are listed below:

1. The manuscript would benefit from adding a section on the importance of studying these interactions to the broader audience. Some points perhaps are for better understanding of signaling networks for development of targeted therapies, understanding the evolution of viruses and other pathogens in the context of host-protein interactions, and the importance of selection in immune pathways.

2. Minor modifications:

a. Line 108, Zarrinpar et al 2003 studied the impact of the mutants in vivo, as mentioned later in the paragraph starting at line 566 in the discussion. They used growth assays in hyper-osmolar conditions to study the effects of introduced mutations in vivo.

b. Line 368, 'tied to changes in interaction strength with Pbs2' should be 'tied to changes in interaction strength with Sho1'.

c. Figure 3c, the x-axis label 'Log2FC interaction score compared to WT Pbs2' could be changed to 'Log2FC interaction score with a given SH3 domain containing protein compared to WT Pbs2'.

Reviewer #3 (Comments for the Authors (Required)):

Jordan et al. describe the use of Deep Mutational Scanning (DMS) and DHFR Protein Complementation Fragment assays (PCA) in yeast to explore possible trade-offs between affinity and specificity for short linear motifs (SLiMs) binding. It has been postulated that such interactions may be weak in part due to this tradeoff. The authors use the well-studied proline-rich SLiM in Pbs2 binding to the SH3 domain of Sho1 in *Saccharomyces cerevisiae* as a case study. Their DMS approach allows them to measure the effect of all point mutations across the core Pbs2 class I polyproline motif and its N- and C-terminal flanking regions for binding to Sho1 (via the SH3 domain), as well as the effect of these mutations on binding to the additional 23 SH3 domain-containing proteins in yeast. The authors' assay enables the analysis of interactions between full-length proteins in their native cellular context. The major claims of the paper are that the mutations that most affect affinity for Sho1, which lie in the region immediately surrounding the Pbs2 SH3-binding motif, have little impact on binding to other SH3 domains and that, to the extent that they do, these effects do not correlate with effects on binding to Sho1. This is an interesting result if true, as it demonstrates a way for biology to decouple affinity from specificity and supports that the flanking regions of motifs - known to be important but not well understood - provide an opportunity to tune specificity among paralogs, albeit one that appears to be infrequently used (at least for this system). However, the specificity claim is not well established experimentally. The cell-based DHFR-PCA competition assay is well validated for comparing Pbs2 variant interactions with Sho1, and the interaction scores reported are in general agreement with previously published data. However, the claims surrounding the effects of mutations on specificity are not well validated and some of the analyses are difficult to interpret as presented.

Major points:

The assay used to test interactions with 23 SH3 domains, to profile specificity, is not as well validated as the liquid culture competition assay, which was done multiple times, in different ways, and with a control for variation in expression levels (using interaction with Hog1). The solid media assay relied on colony size as a readout, and this is a concern because the authors demonstrate a growth defect for mutant I87W, the single mutant that shows the greatest change in specificity and that is most important to the claims of the paper. There was no expression control in the solid-media growth assay, nor were replicates reported. The significance of the changes observed is not clear (it is not clear whether or not the many mutations with log₂-fold-change of <<1, in Figure 3C, are significant and replicable).

The extent of interaction of native Pbs2 with other SH3 domains in yeast needs to be discussed. Pbs2 has been reported to interact specifically with Sho1 and not other SH3 domains (Zarrinpar et al.; 2003). Yet in this paper, the authors report mutations that reduce the binding of Pbs2 to SH3 domains BBC1 and YSC84 and, perhaps not significantly, to 10 other SH3 domains. Was there a strong enough positive signal for these other domains to detect decreases in interaction strength? If so, this is different from the previous reports of low off-target binding and deserves comment. In the Discussion, it is noted that Zarrinpar et al. reported mutations that increase Pbs2 cross-talk, whereas this study finds "that most mutants have specificity comparable to wild-type Pbs2." However, if this study detects interaction of wild-type Pbs2 with other SH3 domains (from which decreases can be quantified), that is a different reference point, which should be discussed. In summary, the authors do not address the baseline interaction of native Pbs2 with other domains in their assay, making it hard to assess whether the reported changes in growth are significant (and, as stated above, whether they reflect changes in interaction strengths).

We are left unconvinced, by the data included, that the specificity assay accurately reported on changes in the interaction affinity of Pbs2 mutants with different SH3 proteins. Our concern is heightened by the result that the largest change in binding to a non-Sho1 SH3 domain was a reduction in binding of the 87W variant, relative to native (87I), to BBC1 and YSC84. Position 87 is 6 residues N-terminal of the SH3-binding motif, and it is surprising, from a structural perspective, that this would weaken binding to these domains. The SLiM literature frequently shows mutations to Trp increasing affinity, for many types of domains, and the suggestion of an opposite effect in a (possibly flexible) flanking region requires more supporting evidence. This is especially true because the 2nd major claim of the paper - that affinity of mutants for Sho1 is not correlated with affinity for other SH3 domains, rests almost entirely on I87W (as illustrated in Figure 3C).

Another important point is the unconventional definition of specificity presented by the authors. In this work, specificity is defined as the affinity of the mutant Pbs2 motif for the off-target SH3 domain. However, specificity usually refers to a difference in binding energies, e.g., to ΔG , the difference in the Gibbs free energy of binding between the target (Sho1 SH3) and off-target domains. Or, sometimes specificity is defined as the difference in binding to a target relative to all other off-targets. However, it is defined as a difference, not a single affinity value. By the standard definitions, in this paper, the only mutations for which it is demonstrated that there is a change in `_specificity_` are those where a mutation gives effects in the opposite direction, relative to native, for Sho1 vs. another SH3 domain. This should be explained.

Other points:

The Sho1-Pbs2 interaction score is incorrectly (?) defined in the text as log₂(change in variant frequencies) when in fact it seems to be normalized relative to native. In some figures, the scores are centered on zero and in other cases on 1. Clear and precise definitions, with consistent names and usage, are required.

The sequence similarity of the various SH3 domains should be reported, including similarity in the different parts of the Pbs2 binding site that are relevant, and the results should be discussed in this context.

It is not clear why the mutations that increase Sho2 affinity the most (in Figures S1, S6) were not selected, as these would be most informative for the question being explored (e.g. V91M or V91L, Q89E or Q89D, H86S or H86T or H86D, S83W).

The extended descriptions of the many different versions of the Sho1-Pbs2 assay in the main text makes it somewhat difficult to focus on the main findings regarding affinity and specificity.

Text/figure remarks:

In line 102, the phrasing of "other components of proteins" is unclear.

In line 205, referencing Figure 2A, the phrasing is unclear. The core motif is residues 93-99 but the text reads as though the

short, flanking adjacent section are residues 93-99.

In line 211, referencing Figure 2A, position 100 does not appear to have a strong impact on binding, raising questions as to why it is included in the flanking motif.

In lines 238-9, the difference between the extended motif and flanking sequence is unclear. The use of "flanking residues" and "non-core motif" (line 591) should be better clarified throughout the entirety of the paper. The paper has an overall tendency to use project-specific jargon, including for the different libraries, which should be clarified.

In all structure figures, it would be helpful to orient the reader to the N- and C- terminus of the predicted Pbs2 motif.

For the AF2 analysis of I87W, the authors describe a H-bond driving the interaction of I87W with the Sho1 SH3 domain, yet given their results in 4D, it seems more likely that hydrophobic interactions that are providing these favorable interactions, not the hydrogen bonding.

Figure S2 should have the points distinguished between with or without sorbitol, and with or without MTX.

Figure S8 has a typo with respect to the extended motif being on the right (when it is on the left).

The axes labels are difficult to read due to their small size.

Reviewer #1 (Comments for the Authors (Required)):

Jordan et al. perform deep-mutational scanning of a SH3-binding motif, along with its surrounding regions, and ask whether the mutations affect affinity to its cognate SH3-domain and to other SH3 domains. They use a well-validated fragment-complementation assay to form their conclusions. They also look briefly at protein-structural predictions to make sense of their results. Overall, I believe the experimental work to be of high quality and the results to be of sufficient interest to readers. However, I am not sure I fully agree with the interpretations of all their experiments (see point #2). I would also have liked to see some evolutionary analysis of the implications of their results, given the fact that there are many SH3-binding motifs known to interact with SH3 domains. There is also room to analyze more thoroughly the structural bindings of their SH3 peptide to other SH3 domains. Here are some suggested points to address:

We are glad that the reviewer judges our experimental work to be of high quality, and that our findings are interesting. We also thank the reviewer for their thoughtful analysis of the paper. Based on the comments of multiple reviewers, including this one, we have substantially reworked our manuscript. In particular, we have removed the section pertaining to specificity (comparing WT Pbs2 binding to all yeast SH3 to mutant Pbs2 binding to the same SH3 domains). We have instead replaced it by a section comparing the Sho1-Pbs2 interaction to structures of other SH3 domains in a structure with a long peptide. We subsequently substantially revised the introduction and discussion to reflect this change. We have also addressed the other comments, as detailed below.

1) The authors perform the experiment twice with slight modifications (comparing the full vs the extended motif). The reasons for this are well-motivated and in general the results between both experiments agree. There are exceptions, however, specifically in the regions around the KLPPLP motif (position 86 to 89). Those change from -ve to +ve within a screen. What is the explanation for this? Is this due to some normalization to the mean instead of to the mean of WT sequences? I don't see how an extra copy of SHO1 can make these regions deleterious in the initial screen and I don't understand why the WT sequences don't all have an interaction score of 0 in the repeat experiment. I see that the interaction 'score' is assay dependent, as it depends on what else is being measured. I find this score makes it difficult to compare all the different assays. Perhaps consider normalizing to the WT sequence instead of the average sequence.

As the interaction scores were initially calculated simply as the log-2-fold change, without normalization, interpretation was complex, and scores had to be compared to the score of the wild-type score of that particular screen. Depending on the score of the wild-type sequence in each screen, a deleterious mutation could be above 0 (but below the wild-

type score) in one screen, but below 0 in another. To simplify the interpretation of our results, we have now normalized the results of our interaction screens. The values of the wild-type mutants are rescaled to 0 and the median of nonsense mutants is rescaled to -1, in each screen. Mutants which are beneficial (to binding) now always have positive scores, while mutants which are deleterious (to binding) now always have negative scores. We have added the following lines explaining this normalization:

“The resulting interaction score was calculated as the normalization of log-2-fold change of variant frequencies before and after selection for the interaction using MTX, as detected by targeted sequencing of the *PBS2* locus. Interaction scores were normalized so that scores of 0 are equivalent to wild-type Pbs2, while interaction scores of -1 correspond to the median of all nonsense mutants. Mutants which possess a higher interaction strength than wild-type Pbs2 therefore have a positive interaction score, while mutants which possess a lower interaction score than wild-type Pbs2 have a negative interaction score.” (lines 200-207)

2) I'm not sure if I fully agree with the analysis of changes in specificity. The PCA assay measures affinity, and the Pbs2_Q82W mutant increases its affinity to SH01: its "interaction score" is >1. That same mutant has an "interaction score" with Fus1, Cdc25 and SDC25_17W of less than 0.5. So, it seems the mutant's affinity to those SH3 domains have increased (which the authors claim is a loss of specificity), but the affinity to those domains has increased correspondingly less than the increased in affinity to SH01. For me, this is still potentially an increase in specificity and it might change how one discusses the story of adaptation with respect to specificity/affinity. That is, specificity should be something about the balance in the changes in affinity between the 'true' target.

There are some other potentially interesting questions with the domains that have 'off-target' interactions lost/gained with some mutations. Some domains appear multiple times, is there any reasons? Are these domains more similar/dissimilar to SH01's SH3 domain?

Due to these comments and especially comments from another reviewer, we have decided to remove the section concerning specificity, and Pbs2 variants binding to SH3-containing proteins from the manuscript (everything that was in the section subtitled: Few mutations change the binding strength of Pbs2 to SH3 containing proteins other than Sho1). We therefore did not address this comment. Instead, we have added a section measuring wild-type Pbs2 binding to yeast SH3-containing proteins (subtitled: Comparison of Sho1-Pbs2 binding to other SH3-motif pairs). We also compare the predicted Sho1-Pbs2 interaction to 3 experimentally determined SH3-motif interactions, and propose mechanisms through which Sho1-Pbs2 binding specificity could be maintained.

3) Only 2/10 mutations tested are inside the core KPLPPLP motif (position 93 to 99). I don't know if there is statistical evidence for the statement that there is a position dependent effect.

As mentioned above, we have removed this section from the manuscript, and therefore did not address this comment.

4) The interaction scores should be provided as a supplementary table. The interaction scores were included in the linked Dryad repository. We have now added them as supplementary tables as well, in supplementary tables S1, S2 and S3.

5) I am not a seasoned structural biologist, but the structure for the Pbs2 extended sequence motif is shown in a strange orientation. I can't fully grasp how even a loop region could fold this way. Currently, it looks like 5-10 amino acids are tangled in a knot, which I would argue is not likely to be a sensible AlphaFold prediction. This poses some questions because the extended motif does not really have any amino acids that increase/decrease interactions but it does look like it forms extended contacts with the structure (as portrayed).

After reviewing the structure in a visualization tool, it seems like the “knot” is simply a visual effect based on the angle at which the structure is viewed. We have changed all visualizations in the paper to show Pbs2 more clearly. It is true that there seems to be contacts between Pbs2 in positions following the motif, but that these positions do not show major effects when mutated. We have added the following lines to the manuscript, addressing this:

“Certain Pbs2 residues towards the C-terminus relative to the canonical motif are also predicted to be in proximity to Sho1, with 2 residues within 2 Å of Sho1, but as mutations in these positions did not have substantial effects on binding (Figure 2a), we decided to concentrate on residues in and near the extended motif.” (lines 353-356)

6) Some of the stories on evolution of motifs are compelling, but I found myself needing to look at alignments to see if this is born out. Indeed, the Pbs2 motif has strong evolutionary conservation in the extended motif. I89, for example, is barely impacted by the mutations on the PCA but is much more conserved than I87 where mutations to hydrophobic amino acids are observed. Why is that? Are other SH3-binding motifs also using the N-terminus of their motif to 'extend' specificity? There is some suggestions that this might be the case, at least with the SH3 'domain-ome' experiments where some mutations in the extended motifs alter binding to other SH3 domains.

Searching in the literature, we have found 3 other examples of yeast SH3 domains interacting with “extended” motifs, as we call them: Abp1-Ark1, Bem1-Ste20 and Nbp2-Ste20. We compare the contacts in these structures to the contacts in the predicted Sho1-Pbs2 complex, and propose ways in which the differences in contacts could allow these extended motifs to modulate specificity. In the case of position 87 and 89, it seems that position 87 is important for Pbs2 binding to Sho1, while position 89 (Q89) has its side-chain pointed away from Sho1. We therefore hypothesize that it does not participate in binding, and is therefore tolerant to mutations (at least for affecting binding strength). We are not certain why it is conserved, but this could be because of some other function fulfilled by Pbs2.

7) Figure 4 B/C should be revised to have some form of concordance on the directions for 'interactions gained/lost'.

This figure was removed from the revised manuscript. Consequently, this comment could not be addressed.

Minor comments:

1) Is the YPD really made with 2% Tryptone and not Peptone?

Tryptone is a type of peptone. Peptones refer to all protein digests, while tryptone is a casein digested by trypsin. See this brochure for detailed information: [https://www.carlroth.com/medias/Info-Brochure-Peptones-EN.pdf?context=bWFzdGVyfHRIY2huaWNhbERvY3VtZW50c3wyMDA5NzN8YXBwbGljYXRpb24vcGRmfHRIY2huaWNhbERvY3VtZW50cy9oMjYvaDI2LzlxMTY3Mzk0MDM4MDYucGRmfGRINmExMmQyMwVhMmJjZTI5ZGE2ZmViYThhMGU0YzMONWRkMDQzMjU4Y2Y4MGQ5ZmU1MjNjM2Q4YjJhNjA2Yzg#:~:text=The%20term%20peptone%20is%20used,pancreas%20extract%20\(pancreatic%20digest\).](https://www.carlroth.com/medias/Info-Brochure-Peptones-EN.pdf?context=bWFzdGVyfHRIY2huaWNhbERvY3VtZW50c3wyMDA5NzN8YXBwbGljYXRpb24vcGRmfHRIY2huaWNhbERvY3VtZW50cy9oMjYvaDI2LzlxMTY3Mzk0MDM4MDYucGRmfGRINmExMmQyMwVhMmJjZTI5ZGE2ZmViYThhMGU0YzMONWRkMDQzMjU4Y2Y4MGQ5ZmU1MjNjM2Q4YjJhNjA2Yzg#:~:text=The%20term%20peptone%20is%20used,pancreas%20extract%20(pancreatic%20digest).)

2) Line 296: small typo (WHen instead of When).

The relevant sentence was removed from the manuscript while rewriting.

Reviewer #2 (Comments for the Authors (Required)):

In this manuscript, the authors investigate the long-suggested trade-off between affinity and specificity in protein-protein interactions in complex signaling pathways. They show through a series of mutational experiments, that regions outside the core motif in Pbs2 contribute to affinity to SH3 domain of Sho1, and labelled them extended motif. They showed that the majority of residues in the core binding motif are already optimized for binding to SH3, whereas residues in the rest of the extended motif can be mutated to improve affinity. They followed this with mutational scanning to investigate whether the improved or reduced affinity mutations introduced in the extended motif, have an opposing effect on specificity, and found no correlation between gain of affinity and loss of specificity. They finally use computational protein modelling to investigate some of the improved affinity mutations, and found that increased hydrophobicity in certain residues in the core domain motif allow better interaction with the hydrophobic pocket formed in the SH3 and adjacent region. Overall, the manuscript is well-written, and experiments are well performed and interpreted. Few suggestions are listed below:

We thank the reviewer for their comments, and are glad that they thought the manuscript was well-written. Based on the comments of the other two reviewers, we have substantially reworked our manuscript. In particular, we have removed the section pertaining to specificity (comparing WT Pbs2 binding to all yeast SH3 to mutant Pbs2 binding to the same SH3 domains). We have instead replaced it by a

section comparing the Sho1-Pbs2 interaction to structures of other SH3 domains in a structure with a long peptide. We subsequently substantially revised the introduction and discussion to reflect this change. We have also addressed the other comments, as detailed below.

1. The manuscript would benefit from adding a section on the importance of studying these interactions to the broader audience. Some points perhaps are for better understanding of signaling networks for development of targeted therapies, understanding the evolution of viruses and other pathogens in the context of host-protein interactions, and the importance of selection in immune pathways.

We have added some lines to explain the impacts of motifs in health and biotechnology. The second paragraph of the introduction now reads:

“Many interaction domains bind short intrinsically disordered stretches of their interaction partners, also known as short linear motifs (SLiMs) or simply binding motifs (Gouw *et al.* 2018). Binding motifs are involved in a wide swath of cell signaling pathways, and underlie many important mechanisms in human and other cells (Kumar *et al.* 2024). Uncovering the determinants of binding can elucidate the functioning of human cells and certain diseases that affect them, such as various cancers in which motif-mediated signaling plays a part (Van Roey *et al.* 2014). Additionally, viruses also use their own binding motifs to disrupt the signaling machinery of their hosts (Davey *et al.* 2011). Understanding domain-motif binding can also allow them to be used as tools, for example by attaching therapeutic proteins to domains and motifs to hydrogels, to ensure the gradual release of the therapeutic proteins in human tissues (Delplace *et al.* 2019).” (lines 84-93)

2. Minor modifications:

a. Line 108, Zarrinpar *et al.* 2003 studied the impact of the mutants *in vivo*, as mentioned later in the paragraph starting at line 566 in the discussion. They used growth assays in hyper-osmolar conditions to study the effects of introduced mutations *in vivo*.

Zarrinpar *et al.* did measure the growth of 3 Pbs2 variants in 4 different conditions (Figure 4f in Zarrinpar *et al.* 2003), but did not measure protein-protein interaction strength *in vivo*. We want to show how our project is a reexamination of many of the questions they previously studied, using the high-throughput methods which are available to us 21 years later. To this end, we have rewritten the beginning of the fourth paragraph of the introduction as follows:

“Binding affinity and specificity have previously been explored in the context of the interaction between binding domains and motifs, by measuring the binding of mutants *in vitro* (Yu *et al.* 1994; Zarrinpar *et al.* 2003; Tonikian *et al.* 2009; Vincentelli *et al.* 2015; Kazlauskas *et al.* 2016). However, the interaction of proteins in the cellular environment is a more complex situation, and many interactions detected *in vivo* are not detected *in vitro*, and vice versa (Kelil *et al.* 2017). Many factors can modulate binding in the cell, including colocalization of partners, expression in the same cell cycle phases, and contributions from sequences outside of the immediate binding domain and motif, as well as other proteins that

can interact with one or both of the partners (Ivarsson and Jemth 2019; Dionne et al. 2022). Earlier studies have measured the effects of limited numbers of domain and motif mutants *in vivo* (Zarrinpar et al. 2003; Marles et al. 2004). More recently, techniques such as deep mutational scanning (DMS) have been used to study the impact of large libraries of mutations on protein stability and function *in vivo* (Fowler and Fields 2014).” (lines 109-121)

b. Line 368, 'tied to changes in interaction strength with Pbs2' should be 'tied to changes in interaction strength with Sho1'.

Due to comments from another reviewer, we have decided to remove the section concerning specificity, and Pbs2 variants binding to SH3-containing proteins from the manuscript (everything that was in the section subtitled: Few mutations change the binding strength of Pbs2 to SH3 containing proteins other than Sho1). We therefore did not address this comment, as we have removed these lines from the text

c. Figure 3c, the x-axis label 'Log2FC interaction score compared to WT Pbs2' could be changed to 'Log2FC interaction score with a given SH3 domain containing protein compared to WT Pbs2'.

As mentioned above, we have removed the section of the manuscript dealing with specificity, and have removed figure 3. We therefore cannot address this comment.

Reviewer #3 (Comments for the Authors (Required)):

Jordan et al. describe the use of Deep Mutational Scanning (DMS) and DHFR Protein Complementation Fragment assays (PCA) in yeast to explore possible trade-offs between affinity and specificity for short linear motifs (SLiMs) binding. It has been postulated that such interactions may be weak in part due to this tradeoff. The authors use the well-studied proline-rich SLiM in Pbs2 binding to the SH3 domain of Sho1 in *Saccharomyces cerevisiae* as a case study. Their DMS approach allows them to measure the effect of all point mutations across the core Pbs2 class I polyproline motif and its N- and C-terminal flanking regions for binding to Sho1 (via the SH3 domain), as well as the effect of these mutations on binding to the additional 23 SH3 domain-containing proteins in yeast. The authors' assay enables the analysis of interactions between full-length proteins in their native cellular context. The major claims of the paper are that the mutations that most affect affinity for Sho1, which lie in the region immediately surrounding the Pbs2 SH3-binding motif, have little impact on binding to other SH3 domains and that, to the extent that they do, these effects do not correlate with effects on binding to Sho1. This is an interesting result if true, as it demonstrates a way for biology to decouple affinity from specificity and supports that the flanking regions of motifs - known to be important but not well understood -

provide an opportunity to tune specificity among paralogs, albeit one that appears to be infrequently used (at least for this system). However, the specificity claim is not well established experimentally. The cell-based DHFR-PCA competition assay is well validated for comparing Pbs2 variant interactions with Sho1, and the interaction scores reported are in general agreement with previously published data. However, the claims surrounding the effects of mutations on specificity are not well validated and some of the analyses are difficult to interpret as presented.

We thank the reviewer for their thoughtful analysis of the paper. Based on the comments of multiple reviewers, including this one, we have substantially reworked our manuscript. In particular, we have removed the section pertaining to specificity (comparing WT Pbs2 binding to all yeast SH3 to mutant Pbs2 binding to the same SH3 domains, i.e., everything that was in the section subtitled: Few mutations change the binding strength of Pbs2 to SH3 containing proteins other than Sho1). We have instead replaced it by a section comparing the Sho1-Pbs2 interaction to structures of other SH3 domains interacting with a long peptide (subtitled: Comparison of Sho1-Pbs2 binding to other SH3-motif pairs). We subsequently substantially revised the introduction and discussion to reflect this change. We have also addressed the other comments, as detailed below.

Major points:

1. The assay used to test interactions with 23 SH3 domains, to profile specificity, is not as well validated as the liquid culture competition assay, which was done multiple times, in different ways, and with a control for variation in expression levels (using interaction with Hog1). The solid media assay relied on colony size as a readout, and this is a concern because the authors demonstrate a growth defect for mutant I87W, the single mutant that shows the greatest change in specificity and that is most important to the claims of the paper. There was no expression control in the solid-media growth assay, nor were replicates reported. The significance of the changes observed is not clear (it is not clear whether or not the many mutations with log₂-fold-change of <<1, in Figure 3C, are significant and replicable).

The extent of interaction of native Pbs2 with other SH3 domains in yeast needs to be discussed. Pbs2 has been reported to interact specifically with Sho1 and not other SH3 domains (Zarrinpar et al.; 2003). Yet in this paper, the authors report mutations that reduce the binding of Pbs2 to SH3 domains BBC1 and YSC84 and, perhaps not significantly, to 10 other SH3 domains. Was there a strong enough positive signal for these other domains to detect decreases in interaction strength? If so, this is different from the previous reports of low off-target binding and deserves comment. In the Discussion, it is noted that Zarrinpar et al. reported mutations that increase Pbs2 cross-talk, whereas this study finds "that most mutants have specificity comparable to wild-type Pbs2." However, if this study detects interaction of wild-type Pbs2 with other SH3 domains (from which decreases can be quantified), that is a different reference point, which should be discussed. In summary, the authors do not address the baseline interaction of native Pbs2 with other domains in their assay, making it hard to assess whether the reported changes in growth are significant (and, as stated above, whether they reflect changes in interaction strengths).

We are left unconvinced, by the data included, that the specificity assay accurately reported on changes in the interaction affinity of Pbs2 mutants with different SH3 proteins. Our concern is heightened by the result that the largest change in binding to a non-Sho1 SH3 domain was a reduction in binding of the 87W variant, relative to native (87I), to BBC1 and YSC84. Position 87 is 6 residues N-terminal of the SH3-binding motif, and it is surprising, from a structural perspective, that this would weaken binding to these domains. The SLiM literature frequently shows mutations to Trp increasing affinity, for many types of domains, and the suggestion of an opposite effect in a (possibly flexible) flanking region requires more supporting evidence. This is especially true because the 2nd major claim of the paper - that affinity of mutants for Sho1 is not correlated with affinity for other SH3 domains, rests almost entirely on I87W (as illustrated in Figure 3C).

The reviewer raises very good points, and has caused us to reconsider our assay. The main problem with the specificity assay is that we are mostly measuring spurious interactions between Pbs2 and various SH3-containing proteins. Weakening of spurious interactions (which we class as gain of specificity) or small strengthening of spurious interactions (which we class as loss of specificity) still result in spurious interactions, and we cannot be assured of having the appropriate resolution to define this. We have decided to change our approach in looking at specificity, and only consider the binding of wild-type Pbs2 (and a control where the motif is replaced by a neutral stuffer sequence). We compare the binding of these two Pbs2 variants to all yeast SH3-containing proteins, and find that the Pbs2 motif interacts specifically with Sho1 *in vivo*. We have taken a more qualitative approach to studying specificity. We have found three other examples of yeast SH3 domains binding “extended motifs” as we call them. We compare the predicted Sho1-Pbs2 structure to these three other structures (Abp1-Ark1, Nbp2-Ste20 and Bem1-Ste20), and compare the contacts made by the motifs in the 4 cases. We propose differences in binding, which may explain how extended motifs can modulate specificity. We no longer make any claims of correlation (or lack thereof) between specificity and affinity. The new analyses can be found in the section subtitled: Comparison of Sho1-Pbs2 binding to other SH3-motif pairs.

2. Another important point is the unconventional definition of specificity presented by the authors. In this work, specificity is defined as the affinity of the mutant Pbs2 motif for the off-target SH3 domain. However, specificity usually refers to a difference in binding energies, e.g., to ΔG , the difference in the Gibbs free energy of binding between the target (Sho1 SH3) and off-target domains. Or, sometimes specificity is defined as the difference in binding to a target relative to all other off-targets. However, it is defined as a difference, not a single affinity value. By the standard definitions, in this paper, the only mutations for which it is demonstrated that there is a change in `_specificity_` are those where a mutation gives effects in the opposite direction, relative to native, for Sho1 vs. another SH3 domain. This should be explained.

As previously mentioned, we have removed the section pertaining to specificity of Pbs2 mutants in our manuscript. We therefore cannot address this comment.

Other points:

3. The Sho1-Pbs2 interaction score is incorrectly (?) defined in the text as $\log_2(\text{change in variant frequencies})$ when in fact it seems to be normalized relative to native. In some figures, the scores are centered on zero and in other cases on 1. Clear and precise definitions, with consistent names and usage, are required.

The Sho1-Pbs2 interaction was correctly defined as $\log_2(\text{change in variant frequencies})$. In cases where the wild-type variant was more beneficial than the average mutant, the wild-type score was above 0. In cases where the wild-type score was about as beneficial as the average variant, the score was around 0. To clarify the score of the wild-type variant, and to simplify interpretation, we have now additionally normalized the interactions scores, so that wild-type variants have a score of 0, and the median of nonsense variant scores is -1. Variants that interact more strongly than wild-type Pbs2 now have positive scores, while variants that interact less strongly than wild-type Pbs2 have negative scores. This is now clearly and precisely defined in the results section as well as the methods section.

“The resulting interaction score was calculated as the normalization of \log_2 -fold change of variant frequencies before and after selection for the interaction using MTX, as detected by targeted sequencing of the *PBS2* locus. Interaction scores were normalized so that scores of 0 are equivalent to wild-type Pbs2, while interaction scores of -1 correspond to the median of all nonsense mutants. Mutants which possess a higher interaction strength than wild-type Pbs2 therefore have a positive interaction score, while mutants which possess a lower interaction score than wild-type Pbs2 have a negative interaction score.” (lines 200-207)

4. The sequence similarity of the various SH3 domains should be reported, including similarity in the different parts of the Pbs2 binding site that are relevant, and the results should be discussed in this context.

We have added a section and a supplementary figure (Figure S14) where we make a multiple sequence alignment of yeast SH3 domains, and 25 residues on each side. We also make a multiple structural alignment using the MUSTANG tool. We highlight positions which are predicted to be in proximity to Pbs2. We discuss this in the section entitled “Comparison of Sho1-Pbs2 binding to other SH3-motif pairs”. The text is copied below:

“As we propose a role in modulating interaction strength for the extended motif and the additional binding pocket of Sho1, we wondered if these structures could play a part in modulating the binding specificity of Pbs2 to Sho1. If this additional pocket were a unique structure among yeast SH3 domains, that could increase the specificity of binding, as the Pbs2 extended motif could be uniquely tuned to bind this unique feature. First, we generated a multiple sequence alignment of yeast SH3 domain sequences with 25 adjacent residues on each side, to verify the conservation of the amino acids forming the additional binding pocket. This window

includes all residues predicted to be in proximity to Pbs2 in Sho1. Apart from a few key residues in the SH3 domain, we found no significant conservation of the positions which in Sho1 are predicted to be in proximity ($< 5 \text{ \AA}$) to Pbs2 positions situated in the extended motif, but outside of the canonical motif (Figure S14a). We reasoned that the structure of the additional binding pocket might be conserved, even if the particular residues were not, so we used the MUSTANG (Konagurthu *et al.* 2006) tool to carry out a multiple structural alignment on structure predictions of yeast SH3 domains with 25 neighboring residues, obtained from the AlphaFold Protein Structure Database (Varadi *et al.* 2022). We found that only the positions in the SH3 domains were structurally conserved, with no significant structural conservation for the positions outside the SH3 domains (Figure S14b), including those forming the additional binding pocket. We therefore conclude that the additional binding pocket on Sho1 is a unique structural feature.” (lines 418-435)

a

b

Figure S14.

Sequence and structural alignments of yeast SH3 domains and adjacent sequences a) Multiple sequence alignment of yeast SH3 domains with 25 residues on each side. Residues which in the predicted Sho1-Pbs2 structure are within 5 Å of Pbs2 residues which are in the extended motif but not the canonical motif are surrounded by a black rectangle. Beginning and end of the SH3 domain of Sho1 marked above the alignment. b) Multiple structure alignment of predicted yeast SH3 domains with 25 residues on each side, from MUSTANG. Residues which in the predicted Sho1-Pbs2 structure are within 5 Å of Pbs2 residues which are in the extended motif but not the canonical motif are surrounded by a black rectangle. Beginning and end of the SH3 domain of Sho1 marked above the alignment.

(lines 1348-1358)

5. It is not clear why the mutations that increase Sho2 affinity the most (in Figures S1, S6) were not selected, as these would be most informative for the question being explored (e.g., V91M or V91L, Q89E or Q89D, H86S or H86T or H86D, S83W).

We chose the mutants after only the first assay (on the entire region surrounding the Pbs2 motif), and so could only base ourselves on this data. We tried to choose mutants which covered the range of possible interaction scores, including a number of the most strongly interacting mutants. We did include many of the mutants mentioned in this comment. They did not appear in (the previous version of) figure 3c, because that figure only included interactions which were different from WT Pbs2 interactions with the same domain with an FDR corrected p-value below 0.05.

6. The extended descriptions of the many different versions of the Sho1-Pbs2 assay in the main text makes it somewhat difficult to focus on the main findings regarding affinity and specificity.

We have gone through the text, and have cut down the descriptions of the different assays. We hope that the text is now more readable. We have also, as previously described, removed the assay of Pbs2 variant binding to yeast SH3-containing proteins, which considerably lightens the manuscript.

Text/figure remarks:

7. In line 102, the phrasing of "other components of proteins" is unclear.

In rewriting the introduction, we have removed this sentence.

8. In line 205, referencing Figure 2A, the phrasing is unclear. The core motif is residues 93-99 but the text reads as though the short, flanking adjacent section are residues 93-99.

We have changed that section to read as follows:

“Many mutations in the canonical motif change the interaction strength. However, we found that most mutations in the region surrounding the motif did not affect binding of Sho1, apart from a short section neighboring the canonical motif itself (Figure 2a, Figure S1, Figure S2, table S1).” (lines 209-211)

We also explicitly define the positions of the canonical motif later in the paragraph:

“Since the effect on binding was strongly position dependent, we categorized different sections of Pbs2 as follows: the canonical binding motif is the conserved type I binding motif sequence situated in positions 93 to 99, the extended motif is composed of the canonical motif and the neighboring residues which have a strong impact on binding from positions 85 to 99 (Figure 2b).” (lines 214-218)

9. In line 211, referencing Figure 2A, position 100 does not appear to have a strong impact on binding, raising questions as to why it is included in the flanking motif.

We have removed position 100 from our definition of the extended motif. We have added this sentence to explain its inclusion in the second screen (which is supposed to focus on the extended motif only):

“In light of these results, we decided to focus solely on the extended motif of Pbs2, comprising codons 85 to 99, and adding position 100 to verify no effects were taking place in the residue immediately following the canonical motif.” (lines 262-264)

10. In lines 238-9, the difference between the extended motif and flanking sequence is unclear. The use of “flanking residues” and “non-core motif” (line 591) should be better clarified throughout the entirety of the paper. The paper has an overall tendency to use project-specific jargon, including for the different libraries, which should be clarified.

We have read through the paper and tried to remove all jargon. In particular, we have removed all references to the flanking sequences, instead referring to such sequences as being “outside the extended motif”. We have also removed the term non-core motif as well as core motif, and instead use the consistent term canonical binding motif when referring to the conserved binding motif in positions 93 to 99. We have also reviewed the use of the term extended motif, and have clarified instances where it was unclear what positions we were referring to. We feel we have reduced the project-specific terms as much as possible, while keeping the manuscript legible.

11. In all structure figures, it would be helpful to orient the reader to the N- and C-terminus of the predicted Pbs2 motif.

On protein structure figures of the predicted Sho1-Pbs2 interaction, we have added annotations pointing towards the N and C terminus of Pbs2. However, we did not add these annotations to the new figures where we compared Sho1-Pbs2 to other interaction structures, as we felt that this would overload the figures and render them much less clear.

12. For the AF2 analysis of I87W, the authors describe a H-bond driving the interaction of I87W with the Sho1 SH3 domain, yet given their results in 4D, it seems more likely that hydrophobic interactions that are providing these favorable interactions, not the hydrogen bonding.

We have rewritten certain sentences to emphasize the role of the hydrophobic interactions:

“In particular, I87W, which is the strongest interacting mutant measured, positions its side-chain in close contact (1.77 Å) with this non-canonical hydrophobic pocket, and particularly in a sub-pocket formed of both SH3 and non-SH3 residues. Additionally, it is predicted to form a hydrogen bond with the aspartic acid side chain in position 333 of the Sho1 sequence (Figure 3c). These two factors potentially explain its strong interaction strength relative to all other Pbs2 variants.” (lines 360-365)

And later:

“Based on the results of the hydrophobic or non-hydrophobic substitutions, we reasoned that these three positions situate their side chains in the non-canonical hydrophobic binding pocket. This finding helps explain the strong interactions strength of Pbs2 mutant I87W, by showing the important role of hydrophobic interactions in position 87.” (lines 378-381)

13. Figure S2 should have the points distinguished between with or without sorbitol, and with or without MTX.

Supplementary figure 2 was split into 3 figures (Figures S2, S4, S10) to allow for added panels, and now each measure in each screen in each condition is represented by its own point.

14. Figure S8 has a typo with respect to the extended motif being on the right (when it is on the left).

The typo was fixed to refer to the appropriate panels. The legend now reads:

“Results from the DMS library of the extended motif (left) and the DMS library of the surrounding region (right)” (lines 1276-1277 and repeated on lines 1284-1285)

15. The axes labels are difficult to read due to their small size.

We have gone through and increased the axis label size in the main and supplementary figures where possible.

November 26, 2024

GENETICS-2024-307624

Residues Neighboring an SH3-Binding Motif Participate in the Interaction In Vivo

Dear Dr. Jordan:

Two experts in the field have reviewed your manuscript, and I have read it carefully as well. We appreciate your efforts to revise the manuscript, which now is quite different from the original submission. As such, reviewer #3 raises substantial concerns mostly based on the new content.

Given that the first reviewer was satisfied with your handling of their original comments, we can offer one additional opportunity to respond to the remaining comments of Reviewer #3.

Most importantly, this reviewer raises question about the model of Pbs2-Sho1 binding based on the Alpha-fold (AF) modeling. This reviewer requests substitution tests of predicted contact residues in Sho1. I believe that the Sho1 "stuffer" mutation that you made gets part of the way there; however, neither I nor the reviewer could find controls that that mutation does not disrupt Sho1 stability and thus abundance, which would give an artifactual interaction score in the PCA. It seems also to me that this Sho1 "stuffer" mutant is a critical part of model testing and should be in the main part of the manuscript. (Incidentally, I also found the term "stuffer" confusing and cumbersome; I also could not easily find the amino acid sequence of this string, which should be in the main text).

Reviewer #3 also raises several other points, many of which can be addressed by text changes. While description of the various DMS screens could be condensed, I do not feel that the screen data need to be moved from the main text.

Finally, and importantly, I also found the manuscript quite difficult to read and suggest that heavy editing of many sentences is needed. Some examples of challenging-to-read sentences to aid you in your editing:

Line 281 "... few single mutations are able to completely prevent" (suggested rewording to, "few single mutations completely prevented ...")

Line 323: "We found that all mutations .. had no significant effect" (suggested rewording to, "none of the mutations had a significant effect")

Line 425: "... all residues predicted to be in proximity to Pbs2 in Sho1" (is the intended point, "all residues in Sho1 that are predicted to be in proximity to Pbs2?")

We look forward to receiving your revised manuscript. Please let the editorial office know approximately how long you expect to need for revisions.

If I am suitably convinced by your response letter and revisions, I hope not to send the manuscript back out for review.

Upon resubmission, please include:

1. A clean version of your manuscript;
2. A marked version of your manuscript in which you highlight significant revisions carried out in response to the major points raised by the editor/reviewers (track changes is acceptable if preferred);
3. A detailed response to the editor's/reviewer #3's feedback and to the concerns listed above. Please reference line numbers in this response to aid the editor and reviewers.

Additionally, please ensure that your resubmission is formatted for GENETICS

<https://academic.oup.com/genetics/pages/general-instructions>

Follow this link to submit the revised manuscript: Link Not Available

Sincerely,

Audrey Gasch
Senior Editor
GENETICS

Approved by:
Howard Lipshitz

Reviewer #1 :

The authors have adequately addressed all my previous concerns.

Reviewer #3 :

The revised paper by Jordan et al. is a significant re-write and emphasizes different conclusions than the initial submission. The main finding now is that an extended region, N-terminal to the canonical SH3-binding motif in Pbs2, contributes to binding affinity for Sho1. The authors present a structural model that can rationalize this observation. The authors also argue that the extended motif region may provide specificity for Sho1 over other yeast SH3 containing proteins. This result is not unexpected, given what is known in the field, but detailed characterization of a mechanism by which residues outside of the SH3 core motif impact affinity and specificity, in cells, would be welcome. However, there are gaps in the support needed to make this model convincing and the paper is still somewhat difficult to read and follow in its details.

The main line of reasoning behind the key result is:

- Residues in Pbs2 positions 87, 90, 91, 92, N-terminal to the canonical motif, impact Sho1 binding, as measured in high-throughput PCA assays
- An AlphaFold prediction of the structure of the Sho1-Pbs2 complex suggests that this region, and residues 87 and 90 in particular, can engage an interface on Sho1 formed by the SH3 domain and another part of the protein (a nearby loop, I gather - it is hard to tell from Figure 3a what part of Sho1 this is that is shown in yellow; need to give residue numbers)
- Bioinformatics and structural analyses are consistent with the AF-predicted interaction being unique to Sho1, compared to other yeast SH3 proteins

The model is supported by mutations in Pbs2. However, there is no test of whether the regions of Sho1 that are predicted by AF to be essential to the specific Pbs2-Sho1 interaction are important, aside from "Sho1_stuffed". Figures 3 and 5 show specific Sho1 residues in both the SH3 domain and the "loop" region that are predicted to be important (e.g., D333, L290, Y293, W338, Y319, I350). These should be mutated to test for effects on the binding of Pbs2, with appropriate controls to test for disrupting the structure of Sho1.

The one mutation of Sho1 that is presented is the "Sho1_stuffed" construct, in which the entire SH3 domain was replaced by a G/S sequence. This is an extreme perturbation and, very surprisingly, this construct retains interaction with Pbs1 (Figure S15). This seems to contradict - rather than support - the AF model in which the binding pocket for the extended motif region is formed at the interface of the SH3 and loop regions, as shown in Figure 3a. This requires more investigation and explanation.

With respect to the arguments about specificity, use of the Pbs2_stuffed construct suggests that other yeast SH3 domain proteins, Abp1 and Nbp2, interact with Pbs2 via regions distinct from the segment that is studied here (Figure S15). There are interesting questions related to whether it is the Sho1-binding core motif (93-99), or the extended flank (85-92) (or the combination of the two) that make this region of Pbs2 selective for Sho1, but that is not resolved here. Figure 5 shows that other SH3 domains that make contacts with residues N-terminal to the core motif use different strategies than Sho1, but the authors do not use AlphaFold to explore whether favorable contacts might be possible between Pbs2 (71-126) and the full-length SH3 proteins in some other binding mode. Of course, claims about the relevance of the extended motif for specificity ultimately require testing Pbs2 variants for interaction with other SH3 domains.

Other comments:

Lines 234-236 "Strikingly, only mutations in the residues comprising the extended motif had Sho1- interaction specific effects, while mutations outside the extended motif had either little effect or destabilizing effects." This doesn't appear to be supported by the data in Figure 2a, which shows that residues outside of the highlighted blue region have improved interaction scores (e.g. at positions 80, 82, 83 and to a lesser extent at position 73, 102, 107, etc.). If the authors wish to focus on the extended motif region, they can just do so, without using this poorly supported statement as a justification. In fact, it is not clear that the data from the preliminary screen, in Figure 2a, are needed for the main conclusions of this paper. I recommend moving this to the supplement and perhaps replacing 2a with an equivalent figure for the data from the haploid screen (the data that are summarized in Figure 2c but not mapped to the sequence). Focusing the presentation of the experiments on those that relate most closely to testing the proposed model would make the paper more streamlined and easier to read.

More structural detail illustrating the proposed model is needed. Figure 2B does not provide any information different from 3A, and it would be good to include a structural view of the predicted binding interface formed by the linker region / SH3 domain and a visualization (in sticks) that locates residues 87, 90, 91. The identity of the residues in the postulated pocket Sho1 pocket should be given. Higher-resolution information about the confidence of the AlphaFold predictions, e.g. pLDDT and pAE at the residue level in the region of interest, should be reported.

The structures in Fig 5 were not very helpful. It was not clear what we were intended to take away. The figure could be better designed to reinforce key points about how SH3 proteins are different in the key regions for this model, perhaps using a zoomed-out view where the relationships between the extended regions and the different SH3 domains can be visualized.

To us, the term "stuffed" seems overly colloquial and not usefully descriptive.

It would be useful to the reader to label residues every time extended motif (85 - 99) and surrounding region (XX- XX) of the Pbs2 motif are referred to, it gets confusing to read supplemental figure legends referring to extended vs surrounding region, etc.

Figure 5A vs 5B are hard to interpret without a lot of effort. Perhaps color-coding the protein/peptide name labels would help. We don't see the difference between opaque vs. not opaque. Some of the interactions that the authors wish to show are obscured by many overlapping residues. Perhaps hiding hydrogen atoms would help de-clutter the images.

For Figure S13, it is difficult to interpret to what extent the shown residues (H86W, V91L, V91M, and Q89D) engage or do not engage the alternate interface or pocket mentioned. While it is clear the Q89D mutation is pointing away from the Sho1 surface, it would be helpful to modify the representation style, perhaps adding shadows, contacts, stick representations of the Sho1 interacting residues, or anything to provide additional context.

Now that the main conclusions have changed, relative to the initial submission, the paper could be improved by focusing more on the core result, which is somewhat buried in the initial descriptions of the several versions of the high-throughput screens.

Response to reviewers

Reviewer 2

Two experts in the field have reviewed your manuscript, and I have read it carefully as well. We appreciate your efforts to revise the manuscript, which now is quite different from the original submission. As such, reviewer #3 raises substantial concerns mostly based on the new content.

Given that the first reviewer was satisfied with your handling of their original comments, we can offer one additional opportunity to respond to the remaining comments of Reviewer #3.

We thank you for the thoughtful review of our revised manuscript, and for the additional opportunity to respond to comments.

Most importantly, this reviewer raises question about the model of Pbs2-Sho1 binding based on the Alpha-fold (AF) modeling. This reviewer requests substitution tests of predicted contact residues in Sho1. I believe that the Sho1 “stuffer” mutation that you made gets part of the way there; however, neither I nor the reviewer could find controls that that mutation does not disrupt Sho1 stability and thus abundance, which would give an artifactual interaction score in the PCA. It seems also to me that this Sho1 “stuffer” mutant is a critical part of model testing and should be in the main part of the manuscript. (Incidentally, I also found the term “stuffer” confusing and cumbersome; I also could not easily find the amino acid sequence of this string, which should be in the main text).

We have done additional experiments to further explore the Pbs2-Sho1 interaction, and reworked the parts of the manuscript to clarify the points raised. First, we built a series of Sho1 mutants, and measured their interaction strength with Pbs2 (as well as Ybt1 as a control interaction) (paragraph starting at line 614). We also cite a previous paper which showed that the Sho1-stuffer construct is expressed at the same level as Sho1 (Dionne, U. et al. Protein context shapes the specificity of SH3 domain-mediated interactions in vivo. Nat. Commun. 12, 1597 (2021).). We should have mentioned this observation before. We have also rewritten the text to better explain the stuffer and its role, and have added the Sho1-stuffer mutant to figure 4a. We have added the stuffer's amino acid sequence to the main text (line 659).

Reviewer #3 also raises several other points, many of which can be addressed by text changes. While description of the various DMS screens could be condensed, I do not feel that the screen data need to be moved from the main text.

We have cut down the details of the DMS screens in the main text, reserving certain details for the methods section. However, we also feel that the DMS screens have their place in the main text, and have opted to keep the structure of the manuscript as is. As far as we know, this is one of the few examples of DMS of a short linear motif.

Finally, and importantly, I also found the manuscript quite difficult to read and suggest that heavy editing of many sentences is needed. Some examples of challenging-to-read sentences to aid you in your editing:

Line 281 “... few single mutations are able to completely prevent” (suggested rewording to, ‘few single mutations completely prevented ...”

Line 323: “We found that all mutations .. had no significant effect” (suggested rewording to, “none of the mutations had a significant effect”

Line 425: “... all residues predicted to be in proximity to Pbs2 in Sho1” (is the intended point, “all residues in Sho1 that are predicted to be in proximity to Pbs2?)

We have gone through the manuscript paragraph by paragraph, and cut down and simplified overly verbose and complex sentences. As examples, here are the three sentences that were flagged:

Line 281 (now line 252) changed to “The impacts on the Sho1-Pbs2 interaction were varied (Figure 2b, Figure S7 in File S1, Table S2 in File S2). Only a small fraction of mutants exhibited interaction scores comparable to nonsense mutants, indicating that single mutations rarely abolish Sho1-Pbs2 binding completely.”

Line 323 (now line 562) changed to : “With the exception of nonsense mutations, no significant effect on cell proliferation was observed (Welch’s t-test false discovery rate corrected p-value < 0.05)”

Line 425 (now line 638) changed to : “First, we performed a multiple sequence alignment of yeast SH3 domains, including 25 flanking residues on each side, which encompasses all Sho1 residues predicted to be within 5 Å of Pbs2.”

Reviewer 3

The revised paper by Jordan et al. is a significant re-write and emphasizes different conclusions than the initial submission. The main finding now is that an extended region, N-terminal to the canonical SH3-binding motif in Pbs2, contributes to binding

affinity for Sho1. The authors present a structural model that can rationalize this observation. The authors also argue that the extended motif region may provide specificity for Sho1 over other yeast SH3 containing proteins. This result is not unexpected, given what is known in the field, but detailed characterization of a mechanism by which residues outside of the SH3 core motif impact affinity and specificity, in cells, would be welcome. However, there are gaps in the support needed to make this model convincing and the paper is still somewhat difficult to read and follow in its details.

The main line of reasoning behind the key result is:

- Residues in Pbs2 positions 87, 90, 91, 92, N-terminal to the canonical motif, impact Sho1 binding, as measured in high-throughput PCA assays
- An AlphaFold prediction of the structure of the Sho1-Pbs2 complex suggests that this region, and residues 87 and 90 in particular, can engage an interface on Sho1 formed by the SH3 domain and another part of the protein (a nearby loop, I gather - it is hard to tell from Figure 3a what part of Sho1 this is that is shown in yellow; need to give residue numbers)
- Bioinformatics and structural analyses are consistent with the AF-predicted interaction being unique to Sho1, compared to other yeast SH3 proteins

We thank the reviewer for their thoughtful assessment of the revised manuscript. We have added residue numbers to the legend of figure 3a.

The model is supported by mutations in Pbs2. However, there is no test of whether the regions of Sho1 that are predicted by AF to be essential to the specific Pbs2-Sho1 interaction are important, aside from “Sho1_stuffed”. Figures 3 and 5 show specific Sho1 residues in both the SH3 domain and the “loop” region that are predicted to be important (e.g., D333, L290, Y293, W338, Y319, I350). These should be mutated to test for effects on the binding of Pbs2, with appropriate controls to test for disrupting the structure of Sho1. The one mutation of Sho1 that is presented is the “Sho1_stuffed” construct, in which the entire SH3 domain was replaced by a G/S sequence. This is an extreme perturbation and, very surprisingly, this construct retains interaction with Pbs1 (Figure S15). This seems to contradict - rather than support - the AF model in which the binding pocket for the extended motif region is formed at the interface of the SH3 and loop regions, as shown in Figure 3a. This requires more investigation and explanation.

To further investigate the role of non-SH3 surfaces of Sho1 in binding Pbs2, we have built a series of 18 Sho1 mutants. These include all mutations mentioned above. We measured the interaction of these mutants with Pbs2 and Ybt1 (control interaction) using the same DHFR-PCA approach as for the main Pbs2 mutant libraries (paragraph starting at line 614). Certain of these mutations disrupt the Sho1-Pbs2 interaction without disrupting the Sho1-Ybt1 interaction, demonstrating the role of Sho1 residues outside the canonical binding pocket in binding. We did find, however, that

some of the Ybt1 interactions were impacted. Nevertheless, all mutations which disrupted Ybt1 interaction, impacted binding to Pbs2, as expected by our model.

As for the Sho1-stuffer maintaining a certain level of interaction with Pbs2, this can be explained by multiple factors, or a combination thereof. The non-SH3 loop that interacts with the extended motif is still capable of interacting, albeit more weakly, with Pbs2 even in the absence of the SH3 domain. Additionally, the interaction between Sho1 and Pbs2 may be facilitated by the colocalization of the HOG pathway. Indeed, DHFR-PCA can also detect indirect interactions.

With respect to the arguments about specificity, use of the Pbs2_stuffed construct suggests that other yeast SH3 domain proteins, Abp1 and Nbp2, interact with Pbs2 via regions distinct from the segment that is studied here (Figure S15). There are interesting questions related to whether it is the Sho1-binding core motif (93-99), or the extended flank (85-92) (or the combination of the two) that make this region of Pbs2 selective for Sho1, but that is not resolved here.

To measure what portion of Pbs2 is able to bind selectively to Sho1, we expressed 4 fragments of Pbs2 (positions 94-99, positions 93-99, positions 85-100 and positions 69-124) and measured their interaction against the SH3 domains of Sho1, Nbp2 and Abp1, using the DHFR-PCA system (paragraph starting at line 676). This assay shows that the Pbs2(85-100) fragment binds to Sho1 more strongly, and binds Sho1 significantly more strongly than the other SH3 domains, whereas Pbs2(93-99) only binds Sho1 significantly more strongly than Nbp2.

Figure 5 shows that other SH3 domains that make contacts with residues N-terminal to the core motif use different strategies than Sho1, but the authors do not use AlphaFold to explore whether favorable contacts might be possible between Pbs2 (71-126) and the full-length SH3 proteins in some other binding mode. Of course, claims about the relevance of the extended motif for specificity ultimately require testing Pbs2 variants for interaction with other SH3 domains.

As mentioned, testing Pbs2 variant binding to other SH3 containing proteins would be necessary to confirm contacts predicted by AlphaFold. In the first version of this manuscript, we had measurements of this sort. However, the reviewers helped us realize that we were not certain of having a strong enough signal to draw any robust conclusions. For this reason, we decided not to predict Pbs2-other protein complexes, as our experimental framework would not allow us to confirm any predictions.

Other comments:

Lines 234-236 “Strikingly, only mutations in the residues comprising the extended motif had Sho1- interaction specific effects, while mutations outside the extended

motif had either little effect or destabilizing effects.” This doesn’t appear to be supported by the data in Figure 2a, which shows that residues outside of the highlighted blue region have improved interaction scores (e.g. at positions 80, 82, 83 and to a lesser extent at position 73, 102, 107, etc.). If the authors wish to focus on the extended motif region, they can just do so, without using this poorly supported statement as a justification. In fact, it is not clear that the data from the preliminary screen, in Figure 2a, are needed for the main conclusions of this paper. I recommend moving this to the supplement and perhaps replacing 2a with an equivalent figure for the data from the haploid screen (the data that are summarized in Figure 2c but not mapped to the sequence). Focusing the presentation of the experiments on those that relate most closely to testing the proposed model would make the paper more streamlined and easier to read.

We have removed the sentence at lines 234-236, which was supposed to refer to Supplementary figure 3, as its meaning was obviously not clear.

In our opinion as authors, the main conclusions of the paper lie in the narrative of finding interactions in the extended motif, and then exploring why the extended motif has effects on binding. Therefore, we believe the initial screen is an integral part of the manuscript, and kept it as such.

More structural detail illustrating the proposed model is needed. Figure 2B does not provide any information different from 3A, and it would be good to include a structural view of the predicted binding interface formed by the linker region / SH3 domain and a visualization (in sticks) that locates residues 87, 90, 91. The identity of the residues in the postulated pocket Sho1 pocket should be given. Higher-resolution information about the confidence of the AlphaFold predictions, e.g. pLDDT and pAE at the residue level in the region of interest, should be reported.

As figure 2b does not provide any additional data compared to figure 3a, we have removed panel 2b. We have instead included a more detailed view of the predicted Sho1-Pbs2 interaction in panel 3b. This view includes stick visualizations of side chains of interest, and only includes relevant residues, to simplify interpretation. Residues in the extended motif and in the Sho1 binding pocket are identified. Furthermore, we have added 2 plots in supplementary Fig 12c that show residue by residue pLDDT data for the regions of interest for all AlphaFold predictions.

The structures in Fig 5 were not very helpful. It was not clear what we were intended to take away. The figure could be better designed to reinforce key points about how SH3 proteins are different in the key regions for this model, perhaps using a zoomed-out view where the relationships between the extended regions and the different SH3 domains can be visualized.

The panels in what was fig 5 (now supplementary figure 19) were substantially reworked. The view was zoomed out, and only the relevant structures for each example were shown, instead of necessarily showing the 4 structures. The non-relevant residues were made semi-transparent, while the residues of interest were made opaque and outlined. The partners in the structures were also identified with color-coded labels.

To us, the term “stuffed” seems overly colloquial and not usefully descriptive.

We have replaced the term “stuffed” with the phrase “replaced by stuffer sequence”. We have also changed the description of proteins from Protein_stuffed, to protein-stuffer. We have chosen to keep the term stuffer for consistency’s sake, as it has been used by previous papers from other research teams as well as our own going back to 2016 (Biot-Pelletier, D. & Martin, V. J. J. Seamless site-directed mutagenesis of the *Saccharomyces cerevisiae* genome using CRISPR-Cas9. *J. Biol. Eng.* 10, 6 (2016).).

It would be useful to the reader to label residues every time extended motif (85 - 99) and surrounding region (XX- XX) of the Pbs2 motif are referred to, it gets confusing to read supplemental figure legends referring to extended vs surrounding region, etc.

We have added parentheses with positions, as suggested, every time we refer to portions of Pbs2.

Figure 5A vs 5B are hard to interpret without a lot of effort. Perhaps color-coding the protein/peptide name labels would help. We don’t see the difference between opaque vs. not opaque. Some of the interactions that the authors wish to show are obscured by many overlapping residues. Perhaps hiding hydrogen atoms would help de-clutter the images.

As mentioned in response to a comment above, we have substantially reworked figure 5 (which is now supplementary figure 19). We have increased the transparency of non-key residues while keeping key residues opaque. We have also changed the view, and compared only relevant structures instead of all 4 structures. We have color-coded labels indicating the binding partners. We have also hidden hydrogen atoms in the most busy structures.

For Figure S13, it is difficult to interpret to what extent the shown residues (H86W, V91L, V91M, and Q89D) engage or do not engage the alternate interface or pocket mentioned. While it is clear the Q89D mutation is pointing away from the Sho1 surface, it would be helpful to modify the representation style, perhaps adding

shadows, contacts, stick representations of the Sho1 interacting residues, or anything to provide additional context.

We have changed the visualizations in supplementary figure 13 to try and better show the interactions formed. We have made all non-key residues semi-transparent. We show the mutated side chains, as well as the side chains of the residues predicted to be in contact with the mutated residues. We have made all these key residues opaque, outlined, and we have labeled them. Incidentally, we have also modified figure 3c in the same manner.

Now that the main conclusions have changed, relative to the initial submission, the paper could be improved by focusing more on the core result, which is somewhat buried in the initial descriptions of the several versions of the high-throughput screens.

As mentioned above, as authors, we feel that the main conclusion of the paper is the identification of the extended motif and the exploration of some of its properties. As such, we have kept the manuscript in more or less the same form.

June 30, 2025

GENETICS-2025-308278

Residues Neighboring an SH3-Binding Motif Participate in the Interaction In Vivo

Dear Dr. Jordan:

Thank you for the revised submission of your manuscript. I have spent considerable time assessing your revisions so as to avoid sending out for a third round of review. I was convinced by most of the provided revisions with one remaining exception that must be addressed before the manuscript can be accepted for publication in GENETICS.

A crux of the paper is that Sho1 interacts with Pbs2 outside the canonical SH3 domain. I appreciate the addition of new experiments in Figure S14 that test your modeling predictions, by demonstrating the effects of Sho1 substitutions on Pbs1 binding, and on Ybt1 binding as an indirect control for Sho1 stability / abundance.

First, as specified in my last decision, this is now a critical part of the manuscript and should be provided as a main figure. Please move Figure S14 to the main text.

Second, and more importantly, I had some remaining concerns about the data in this figure. Critical to the conclusions are that substitutions in the predicted Pbs1-interacting domain of Sho1 do not affect Sho1's interaction with Ybt1, which happens independently of the SH3 domain. While I appreciate all the work that went into generating these data, one remaining concern is how the Sho1-Ybt1 data are normalized: the replicates show considerably higher variation than the Sho1-Pbs2 data, such that half of the wild-type control measurements are above the normalization level (dashed line) for Ybt1 only. Is it expected that so many of the Sho1 substitutions have higher binding affinity to Ybt1? If not, it raises the possibility that the normalization is off for this important control: indeed, the relative distribution of mutant effects is similar for both Pbs2 and Ybt1 interactions, just shifted higher for Ybt1 across the board. This raises the possibility that the substitutions of interest in this manuscript also affect Ybt1 interactions (or perhaps more likely, the stability of Sho1).

I can see two resolutions: one is to address the data, either by adding more replicates or better providing data for a synonymous substitution to show that that Sho1-Ybt1 interaction fall on the dashed line. Alternatively is to address in text: if there is any question about the normalization, then really only positions 287 and 350 are clearly different for Pbs2, versus Ybt1, binding. I was not convinced by the text on line 624 that states, "We identified four Sho1 positions where mutations specifically weakened the Pbs2 interaction ... [including] positions 319, 333, and 350 ..." Both Y319 substitutions affect the Ybt1 interaction (one fails significance due to wide variance, but it otherwise has a reproducible impact) so that position should be excluded from mention here. If there are questions about the normalization, that raises some issue about position 333. Therefore, qualifications in the text to state that most of these mutations did impact Ybt1 interaction, albeit less-so than the Pbs2 interactions, with the clear exceptions of positions 287 and 350, is required.

A final minor comment: the supplemental methods seem redundant but expanded from the methods in the main text. Please list all critical methods in the main text and provide supplementary methods that expand on data shown on the supplement, where needed. Otherwise, it makes it challenging for readers to find critical information for specific experiments.

I do believe that addressing these changes will clarify and strengthen important points of your work. We look forward to receiving your revised manuscript. Please let the editorial office know approximately how long you expect to need for revisions.

Upon resubmission, please include:

1. A clean version of your manuscript;
2. A marked version of your manuscript in which you highlight significant revisions carried out in response to the major points raised by the editor/reviewers (track changes is acceptable if preferred);
3. A detailed response to the editor's/reviewers' comments and to the concerns listed above. Please reference line numbers in this response to aid the editors.

Additionally, please ensure that your resubmission is formatted for GENETICS.

<https://academic.oup.com/genetics/pages/general-instructions>

Follow this link to submit the revised manuscript: Link Not Available

Sincerely,

Audrey Gasch
Senior Editor

GENETICS

Approved by:
Howard Lipshitz
Editor in Chief
GENETICS

Dear Dr. Jordan:

Thank you for the revised submission of your manuscript. I have spent considerable time assessing your revisions so as to avoid sending out for a third round of review. I was convinced by most of the provided revisions with one remaining exception that must be addressed before the manuscript can be accepted for publication in GENETICS.

We thank you for your multiple evaluations of the different versions of the manuscript. We also thank you for the opportunity to make the changes without a long and protracted third round of revisions.

A crux of the paper is that Sho1 interacts with Pbs2 outside the canonical SH3 domain. I appreciate the addition of new experiments in Figure S14 that test your modeling predictions, by demonstrating the effects of Sho1 substitutions on Pbs1 binding, and on Ybt1 binding as an indirect control for Sho1 stability / abundance.

First, as specified in my last decision, this is now a critical part of the manuscript and should be provided as a main figure. Please move Figure S14 to the main text.

We have added an additional panel to figure 3, with data from certain Sho1 variants. We have done this to highlight the variants of interest, while not overloading the manuscript with data. Figure S14 contains all Sho1 variants, including those present in figure 3.

Second, and more importantly, I had some remaining concerns about the data in this figure. Critical to the conclusions are that substitutions in the predicted Pbs1-interacting domain of Sho1 do not affect Sho1's interaction with Ybt1, which happens independently of the SH3 domain. While I appreciate all the work that went into generating these data, one remaining concern is how the Sho1-Ybt1 data are normalized: the replicates show considerably higher variation than the Sho1-Pbs2 data, such that half of the wild-type control measurements are above the normalization level (dashed line) for Ybt1 only. Is it expected that so many of the Sho1 substitutions have higher binding affinity to Ybt1? If not, it raises the possibility that the normalization is off for this important control: indeed, the relative distribution of mutant effects is similar for both Pbs2 and Ybt1 interactions, just shifted higher for Ybt1 across the board. This raises the possibility that the substitutions of interest in this manuscript also affect Ybt1 interactions (or perhaps more likely, the stability of Sho1).

Normalization was done separately for Pbs2 interactions and Ybt1 interactions. So, all Sho1-Pbs2 interactions are normalized to the median level of the Sho1(WT)-Pbs2 interaction, while all Sho1-Ybt1 interactions are normalized to the median level of the Sho1(WT)-Ybt1 interaction. The Sho1-Ybt1 interaction is naturally weaker than the Sho1-Pbs2 interaction, and so we used this double normalization, because we were interested in the relative change of strength of both interactions. In reality, apart from a few Sho1 mutants that completely killed both interactions (e.g., stuffer, D317I, W338Q), the Sho1(variant)-Pbs2 interaction was always stronger than the same Sho1 variant with Ybt1 (see new version of figure S14 where normalization is not done, included in the manuscript, and copied below for convenience).

Therefore, it is normal that half of the points for the Sho1(WT)-Ybt1 interaction are above the dotted line, as the dotted line is at 1, and the data were all normalized by the median of the Sho1(WT)-Ybt1 interaction.

However, it is true that there seems to be more variability in the Sho1-Ybt1 interactions. As the interaction is weaker, small fluctuations, which are normal in PCA, will be relatively larger when normalized by the wild-type interaction. Furthermore, it seems that Ybt1 is not as good a control of Sho1 stability/abundance as hoped, and not as good as Hog1 was for Pbs2 variants. We do not think that there are any options which will be better. Ybt1 was identified in a previous paper by our lab (Dionne et al 2001, <https://doi.org/10.1038/s41467-021-21873-2>), as an interaction which was as strong with or without the Sho1 SH3 domain. This was identified in a high-throughput screen. Our more precise study of the interaction, and especially the Sho1(stuffer)-Ybt1 interaction, shows that in fact there is a small effect of Sho1(stuffer) on the Ybt1 interaction.

New version of Figure S14. The same data was used, but the normalization of growth rates to the *Sho1*(WT) interaction was not done

I can see two resolutions: one is to address the data, either by adding more replicates or better providing data for a synonymous substitution to show that that *Sho1*-*Ybt1* interaction fall on the dashed line. Alternatively is to address in text: if there is any question about the normalization, then really only positions 287 and 350 are clearly different for *Pbs2*, versus *Ybt1*, binding. I was not convinced by the text on line 624 that states, "We identified four *Sho1* positions where mutations specifically weakened the *Pbs2* interaction ... [including] positions 319, 333, and 350 ..." Both *Y319* substitutions affect the *Ybt1* interaction (one fails significance due to wide variance, but it otherwise has a reproducible impact) so that position should be excluded from mention here. If there are questions about the normalization, that raises some issue about position 333. Therefore, qualifications in the text to state that most of these mutations did impact *Ybt1* interaction, albeit less-so than the *Pbs2* interactions, with the clear exceptions of positions 287 and 350, is required.

These are two good suggestions. Due to the points we raised above, we do not believe that adding replicates or a synonymous *Sho1* variant will significantly change the results. We originally mentioned all the positions in the text where our Student's t-test had shown a significant difference for *Pbs2* but not *Ybt1*. However, we agree that the only two positions with a clear unambiguous signal are 287 and 350. We believe that the signal at position 287 is still enough to show that there is binding outside the SH3 domain.

We have therefore expanded the section of the text discussing the Sho1 mutant growth assay. The discussion is split into two paragraphs. The first paragraph (lines 633 to 644) better explains our approach and what we are testing. The second paragraph (lines 646 to 659) begins by describing the caveats caused by the imperfect Ybt1 control. We then discuss the two clear results: the G287A and I350W mutants.

A final minor comment: the supplemental methods seem redundant but expanded from the methods in the main text. Please list all critical methods in the main text and provide supplementary methods that expand on data shown on the supplement, where needed. Otherwise, it makes it challenging for readers to find critical information for specific experiments.

Following our emails, and the guidance from the Genetics editorial staff, the materials and methods section along with the supplementary methods have been thoroughly reworked. All important details for understanding the paper and recreating the experiments are now found in the main text. The supplementary methods are now much shorter, and contain only complementary information to the main text.

I do believe that addressing these changes will clarify and strengthen important points of your work. We look forward to receiving your revised manuscript. Please let the editorial office know approximately how long you expect to need for revisions.

We thank you for your comments, and we agree that the changes made have improved the manuscript.

July 25, 2025

RE: GENETICS-2025-308419

Mr. David F. Jordan
Universite Laval
Biochimie, microbiologie et bio-informatique
1030, Avenue de la Médecine, Université Laval
Québec, N/A G1V 0A6
Canada

Dear Dr. Jordan:

Congratulations, your manuscript titled "Residues Neighboring an SH3-Binding Motif Participate in the Interaction In Vivo" is accepted for publication in GENETICS! Many thanks for submitting your research to the journal. I appreciate your most recent changes that have addressed my remaining comments.

To Proceed to Publication:

1. Format your article according to GENETICS style: <https://academic.oup.com/genetics/pages/author-guidelines>
2. Ensure that you comply with data and community resource citation guidelines:
<https://academic.oup.com/genetics/pages/author-guidelines#section-5-9-2>
3. Upload your final files at <https://genetics.msubmit.net>
4. Add oupsupport@scipris.com and genetics.oup@novatechset.com (or the domains @scipris.com and @novatechset.com) to your email program's "safe senders" list. You will be contacted by both at various points during the production process.

Notes:

- Your currently-accepted manuscript (unedited, as submitted, reviewed, and accepted) will be published at GENETICS and deposited into PubMed as an Advance Access article. Notify sourcefiles@thegsajournals.org before signing your license if you do not wish to publish your article via Advance Access.
- We invite you to submit an original color figure related to your paper for consideration as cover art. Please email your submission to the editorial office or upload it with your final files. You can submit a small-sized image for evaluation, and if selected, the final image must be a TIFF file 2513px wide by 3263px high (8.375 by 10.875 inches; resolution of 600ppi). Please avoid graphs and small type.
- After files are sent to Oxford University Press we use SciPris to manage article licensing and payment. If you do not have a SciPris account, you will receive an email from no-reply@scipris.com to sign up to use Oxford University Press' author portal. After logging in, follow the online instructions to sign your license and arrange any payment due.

If you have any questions or encounter any problems while uploading your accepted manuscript files, please email the editorial office at sourcefiles@thegsajournals.org.

Sincerely,

Audrey Gasch
Senior Editor
GENETICS

Approved by:
Howard Lipshitz
Editor in Chief
GENETICS